# Bioinformatic Prioritization and Functional Annotation of GWAS-Based Candidate Genes for Primary Open-Angle Glaucoma

**DOI:** 10.3390/genes13061055

**Published:** 2022-06-13

**Authors:** Nigus G. Asefa, Zoha Kamali, Satyajit Pereira, Ahmad Vaez, Nomdo Jansonius, Arthur A. Bergen, Harold Snieder

**Affiliations:** 1Department of Epidemiology, Unit of Genetic Epidemiology and Bioinformatics, University of Groningen, UMCG, Hanzeplein 1, 9713 GZ Groningen, The Netherlands; z.kamali@umcg.nl (Z.K.); a.vaez@umcg.nl (A.V.); h.snieder@umcg.nl (H.S.); 2Department of Bioinformatics, Isfahan University of Medical Sciences, Isfahan P.O. Box 81746-7346, Iran; 3Department of Cell and Molecular Biology, Manipal School of Life Sciences, Manipal Academy of Higher Education, Manipal 576104, India; satyajitdp12@gmail.com; 4Department of Ophthalmology, University of Groningen, UMCG, Hanzeplein 1, 9713 GZ Groningen, The Netherlands; n.m.jansonius@umcg.nl; 5Department of Clinical Genetics, Amsterdam University Medical Center, Meibergdreef 9, 1105 AZ Amsterdam, The Netherlands; aabergen@amsterdamumc.nl

**Keywords:** primary open angle glaucoma, GWAS, DNA methylation, gene expression, functional enrichment

## Abstract

Background: Primary open-angle glaucoma (POAG) is the most prevalent glaucoma subtype, but its exact etiology is still unknown. In this study, we aimed to prioritize the most likely ‘causal’ genes and identify functional characteristics and underlying biological pathways of POAG candidate genes. Methods: We used the results of a large POAG genome-wide association analysis study from GERA and UK Biobank cohorts. First, we performed systematic gene-prioritization analyses based on: (i) nearest genes; (ii) nonsynonymous single-nucleotide polymorphisms; (iii) co-regulation analysis; (iv) transcriptome-wide association studies; and (v) epigenomic data. Next, we performed functional enrichment analyses to find overrepresented functional pathways and tissues. Results: We identified 142 prioritized genes, of which 64 were novel for POAG. *BICC1*, *AFAP1*, and *ABCA1* were the most highly prioritized genes based on four or more lines of evidence. The most significant pathways were related to extracellular matrix turnover, transforming growth factor-β, blood vessel development, and retinoic acid receptor signaling. Ocular tissues such as sclera and trabecular meshwork showed enrichment in prioritized gene expression (>1.5 fold). We found pleiotropy of POAG with intraocular pressure and optic-disc parameters, as well as genetic correlation with hypertension and diabetes-related eye disease. Conclusions: Our findings contribute to a better understanding of the molecular mechanisms underlying glaucoma pathogenesis and have prioritized many novel candidate genes for functional follow-up studies.

## 1. Introduction

The term ‘glaucoma’ refers to a group of ocular disorders characterized by the loss of retinal ganglion cells and the degeneration of their axons [1]. Primary open-angle glaucoma (POAG) is the most common form of glaucoma. While the exact cause of POAG is still unknown, there is clear evidence that age, sex, and intraocular pressure (IOP) are important risk factors. However, genetic factors also play a significant role [1]. Indeed, early evidence from twin and family studies revealed substantial glaucoma heritability [2]. Later on, linkage studies enabled researchers to map the chromosomal locations of a number of rare pathological variants in genes (*MYOC*, *OPTN*, and *WDR36*) that co-segregated with the disease in families [3,4]. More recently, genome-wide association studies (GWASs) have identified a large number of common genomic variants associated with POAG in unrelated individuals. For example, a recent meta-GWAS in 12,315 POAG cases and 227,987 controls, from the Genetic Epidemiology Research on Adult Health and Aging (GERA) cohort and the UK Biobank, identified or replicated more than 70 genomic regions associated with the disease, of which 14 were novel genetic loci [5]. However, a major drawback of GWASs is that the identified variants merely tag genomic regions without providing definitive information on the likely causal genes and functional mechanisms underlying the statistical associations with a particular disease or phenotype [6].

Often, GWASs simply report the nearest genes for each locus. However, in many cases, there are several co-inherited variants in strong linkage disequilibrium (LD) with one another, making it difficult to distinguish the causal variants underlying the statistical association [7]. Moreover, several examples have clearly shown that functionally relevant genes are sometimes located at large distances from the significant GWAS loci [8,9]. This calls for further post-GWAS bioinformatics follow-up studies mapping the identified GWAS associations to the likely causal genes using more robust evidence rather than physical distance to the GWAS loci. Only a limited number of previous studies have attempted to determine the functional characteristics of GWAS-based POAG candidate genes [10,11]. However, a systematic post-GWAS approach that ranks the candidate genes in order of their relevance/causality based on different sources of biological evidence has not been performed yet. Therefore, deep investigations of disease-related genes’ expressions across different tissue types are required to better understand disease mechanisms and to avoid off-target reactions in subsequent therapeutic approaches. 

The relationship between a GWAS association signal, the so-called single-nucleotide polymorphism (SNP), and the related causal gene (variant) is often not clear. An associated SNP may: (i) alter the amino acid coding (i.e., a nonsynonymous SNP (nsSNP)), changing the protein structure and potentially the function of a gene directly, or (ii) may indirectly exert its phenotypic effect through influencing regulatory sequences and, hence, the expression of a gene. Therefore, bioinformatic post-GWAS pipelines will typically check whether GWAS signals will be in high linkage disequilibrium with nonsynonymous SNPs within nearby genes and use publicly available expression quantitative trait loci (eQTL) resources from relevant tissues to check whether associated SNPs in the identified loci overlap with genomic loci related to altered gene expression [12]. Despite being helpful to some extent, finding such an overlap can also be misleading, as it does not provide conclusive evidence that the target gene expression is also associated with the phenotype. There are a number of ways to provide additional information, including transcriptome-wide association studies (TWAS). 

TWAS is a popular approach to test gene expression–phenotype associations. However, this approach does not infer causality, as it is highly vulnerable to environmental confounder effects. Moreover, it requires individual-level phenotype and expression data from the same population, which is hardly ever available on a large scale, limiting the power of detecting true associations. A recently developed new TWAS approach uses previously trained transcriptome models to predict the genetic component of gene expression levels, a method known as transcriptome imputation [13]. This method allows the use of expression and phenotype data from different samples and it concurrently minimizes environmental confounding effects through the use of genetically predicted gene expressions. A recent implementation of this TWAS approach, called MetaXcan, only requires summary statistics to test gene expression associations with the outcome [14,15]. Another summary statistics-based TWAS approach is summary data-based Mendelian randomization (SMR) [16]. This method uses genetic variants as instrumental variables to test for the causative effects of gene expressions (exposure) on disease outcome. We take advantage of the two latter approaches, MetaXcan and SMR, in the current work. Furthermore, given that genetic variants associated with complex diseases, such as POAG, are predominantly located in non-coding regions, there is an existing hypothesis that epigenetic mechanisms (e.g., DNA methylation) mediate the effects of DNA on disease phenotypes via the regulation of gene expression [17,18].

Taken together, in the current post-GWAS study we combine genomic, transcriptomic, and epigenomic data from different sources to elucidate biological mechanisms underlying the pathophysiology of POAG. More specifically, we aimed to (i) prioritize the most likely causal genes and (ii) identify the underlying biological processes and tissues involved in glaucoma through functional enrichment analyses. Furthermore, pleiotropic variants and genetic correlations of POAG with other traits hypothesized to have a relationship with glaucoma (e.g., blood pressure and hypertension [19], type 2 diabetes [20], Alzheimer’s disease [21] and body mass index [22] were also examined). 

## 2. Materials and Methods

We used GWAS summary statistics comprising 12,315 glaucoma cases and 227,987 controls, of which 7329 cases and 169,561 controls were from the UK Biobank, and 4986 cases and 58,426 controls were from the multi-ethnic GERA cohort [5]. We used a post-GWAS bioinformatics-based strategy that follows our previously published pipeline [23] with some modifications. This post-GWAS approach comprises two phases, each of which consists of multiple steps described below (see Figure 1). 

### 2.1. Phase One: Gene Prioritization Analyses

#### 2.1.1. In Silico Sequencing

Of the total of 84 identified or replicated gSNPs by Choquet et al. [5], 75 were selected for further analysis in our study based on the following criteria: (i) discovered in the UK Biobank and replicated (*p* < 0.05) in the GERA cohort or vice versa (*n* = 16), (ii) identified in previous glaucoma GWAS studies and replicated (*p* < 0.05) in the GERA cohort (*n* = 12), and (iii) discovered in the combined systematic meta-analysis (*p* < 5.0 × 10^−8^) of the UK Biobank and GERA cohort (*n* = 47; Table 1). In the case of the occurrence of a gSNP in more than one of the three criteria, the following order of priority of *p*-values was followed—meta-analysis > replication > discovery. The chromosomal locations and alleles of these gSNPs were verified using the Ensembl database and 1000 genomes project phase 3 data reference panel [24]. Next, we clumped the 75 gSNPs based on a physical distance of 1 Mb and *r*^2^ < 0.1 as LD metric, using PLINK (v1.9) [25]. 

The resulting 50 independent gSNPs (i.e., after clumping analysis; Table 1) were used as input for performing in silico sequencing and in silico look-ups of pleiotropic associations with other traits in the GWAS catalog. Using a 1Mb region on either side of the independent top gSNPs, a cut-off value for LD was set at *r*^2^ > 0.50 and the analysis was restricted to the European population. Annotation of the gSNPs together with their linked SNPs was carried out using the ANNOVAR software (version 23rd March 2019) [26]. The possible damaging effects of nonsynonymous SNPs on protein structure and function were predicted using the Sorting Intolerant From Tolerant (SIFT) [27] PROVEAN [28] and Polymorphism Phenotyping (PolyPhen) [29] scoring tools. Furthermore, the pleiotropic effect of POAG-associated loci was assessed using the GWAS catalog database (version 17 March 2019) [30]. 

**Figure 1 genes-13-01055-f001:**
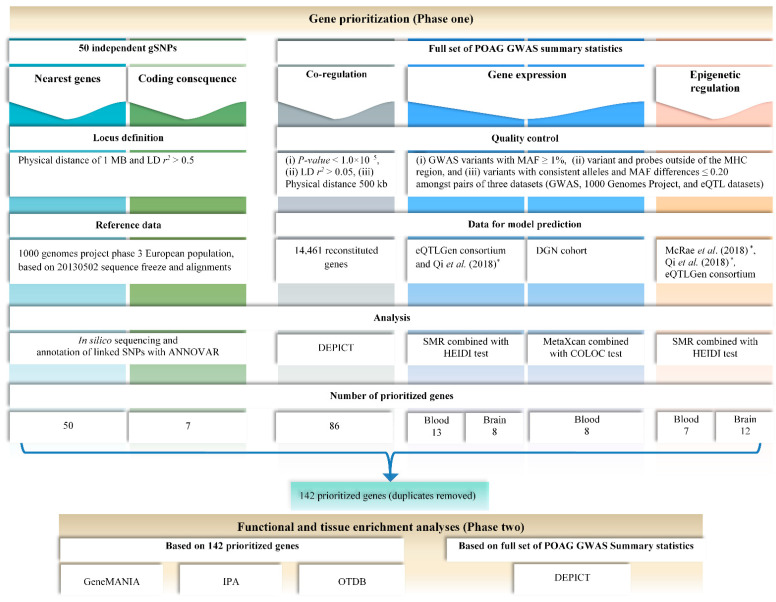
Summary of the analysis pipeline: phase one shows the gene prioritization pipeline which was performed based on (i) 50 independent gSNPs (via nearest genes and coding consequence) and (ii) the full set of POAG GWAS summary statistics, using co-regulation, gene expression, and epigenetic regulation approaches. It also presents a summary of the quality control thresholds, reference panel used, data used for model predictions, method of data analysis, and the final number of genes prioritized for each part separately and for the entire pipeline. Similarly, phase two summarizes the functional and tissue enrichment analyses, which were performed using the 142 prioritized genes and the full set of POAG GWAS summary statistics. * Qi et al. (2018) [31]; and McRae et al. (2018) [32].

#### 2.1.2. Co-Regulated Genes within the POAG-Associated Loci

We ran Data-driven Expression Prioritized Integration for Complex Traits (DEPICT) [33] using the full set of POAG GWAS summary statistics (Figure 1). The significance threshold for index SNPs was set to *p* < 1 × 10^−5^ and clumping was based on *r*^2^ < 0.05 as the LD metric and a physical distance of 500 kb. DEPICT systematically prioritizes the most likely causal genes, gene sets and tissue enrichments based on gene function predictions, even for uncharacterized genes [33]. It performs functional predictions using 14,461 ‘reconstituted gene sets’ based on a curated data set of 77,840 human expression microarrays. DEPICT first identifies all genes at the trait-associated loci and then estimates their co-functionality to prioritize the most likely causal genes [33].

#### 2.1.3. MetaXcan 

In order to identify genes whose expression levels are associated with POAG, free of non-genetic confounders, we performed PrediXcan [13] analysis using the summary data-based pipeline named MetaXcan [15] (Figure 1). This pipeline integrates the summary statistics from genetically predicted transcriptome models with GWAS results to test for the association between the genetic component of gene expression levels and the outcome. We only included GWAS variants with minor allele frequency (MAF) ≥ 1% and, given their high LD ratio, excluding all variants and probes within the major histocompatibility complex (MHC) region. Our analysis was based on the transcriptome model of whole blood from the DGN cohort [34]. The significance level of association *p*-value was set to 4.39 × 10^−6^ regarding 11,397 genes (or variant sets) being tested in the DGN dataset (0.05/11,397 genes). Results were then retained if their prediction performance *p*-value was smaller than a Bonferroni corrected level (0.05/nsign; with nsign being the number of significant genes).

We then tested whether the associated variants for gene expression (eQTLs) and glaucoma are colocalized using the COLOC R package to address LD contamination concerns in significant genes [35]. The test is based on approximate Bayes factors on five hypotheses of (i) no causal variant (H0), (ii) causal variant for glaucoma only (H1), (iii) causal variant for gene expression only (H2), (iv) two distinct causal variants (H3) and (v) single causal variant for gene expression and glaucoma (H4). We took a probability of H3 < 0.5 and H4 > 0.5 as acceptable evidence of colocalized signals to filter out MetaXcan TWAS association results, which can be due to LD, as suggested by Barbeira et al. [15].

#### 2.1.4. Summary Data-Based Mendelian Randomization (SMR) Based on Gene Expression Data

We performed SMR analysis combined with the Heterogeneity Independent Instruments (HEIDI) test (Figure 1), which jointly uses GWAS summary statistics and eQTL data from independent studies; providing more power to detect causal associations [16,36]. In SMR, the top cis-eQTL for each gene is used as ‘instrumental variable’ and ‘gene expression’ is considered the ‘exposure’ for the phenotype. This Mendelian randomization framework enables a test for the causal effect of the genetic variant (i.e., the eQTL) on the phenotype through gene expression [16]. Since the observation of a significant SMR association may be due to two distinct but linked causal variants, one affecting gene expression and the other influencing the phenotype, we also conducted the HEIDI test to ensure that the trait–gene expression associations are driven by the same genetic variant, not confounded by linkage [16].

We used two gene expression datasets: (i) blood eQTL summary data from the eQTLGen consortium (n~32,000) [37] and (ii) a large set of eQTL data from a meta-analysis of 10 brain regions (n effective ~ 1194) [31]. We used the 1000 Genomes Project phase 3 data panel [24] for LD calculations. The following exclusion criteria were applied after which we retained remaining eligible SNPs for the analysis: (i) rare genetic variants with MAF below 1%, (ii) variant and probes within the MHC region, and (iii) variants with inconsistent alleles or MAF differences > 0.20 amongst pairs of three datasets (GWAS, 1000 Genomes Project, and eQTL datasets). Bonferroni’s corrected significance level of *p* < 3.26 × 10^−6^ was set for the blood (0.05/15,352 genes) and *p* < 6.79 × 10^−6^ for the brain (0.05/7361 genes) SMR results. Similarly, a Bonferroni corrected significance level of *p* ≥ 2.78 × 10^−3^ (0.05/18 SMR significant probes) in blood and *p* ≥ 5.0 × 10^−3^ (0.05/10 SMR significant probes) in the brain dataset was set for HEIDI test.

SMR based on methylation data and 3xSMR (DNA→Methylation→Gene expression→POAG).

Given the likely role of epigenetics in complex disease, we also conducted epigenome-wide studies on POAG with SMR using both blood and brain mQTL data (MSMR, see below). That is, the SMR framework affords a test for the causal effect of the genetic variant (i.e., the mQTL) on the phenotype through methylation. We then mapped the significant methylation sites to their nearest genes using the FDb.InfiniumMethylation.hg19 R package [38] (Figure 1).

To further investigate the underlying mediating mechanism from DNA to POAG, we performed 3xSMR analyses on GWAS of POAG using blood and brain data, separately. The 3xSMR analysis was based on three sets of SMR results: (i) GWAS vs. mQTL (MSMR); in which methylation was the exposure and POAG was the outcome, (ii) mQTL vs. eQTL (MESMR); in which methylation was the exposure and gene expression was the outcome, and (iii) GWAS vs. eQTL (ESMR); in which gene expression was the exposure and POAG was the outcome, as described in the section above. The results of these three sets of analyses are integrated all into one causal model [31,36]. This enabled us to identify the associations between DNA, methylation, and gene expression, which consequently may led to the development of POAG (i.e., DNA→Methylation→Gene expression→POAG) (Figure 2).

We used the blood mQTL summary data from a meta-analysis (*n* = 1980) recently reported by Wu et al. [36] and performed MSMR analysis to find the likely causal DNA methylation sites for POAG. Then, to map those methylation sites to the subsequently regulated genes, we repeated SMR analysis on mQTL data [32] using blood eQTL data from the eQTLGen consortium (*n* = 32,000) [37] (MESMR). Finally, we used the blood ESMR results from the previous steps to retain genes with significant causal associations for all three steps (3xSMR).

For MSMR in the brain, we used the brain mQTL summary data from the Qi et al. meta-analysis (estimated effective *n* = 1160) [31] of ROSMAP [39], Hannon et al. [40] and Jaffe et al. [41]. We then performed MESMR using the same brain mQTL data against brain eQTL data reported by Qi et al. [31]. In the final step, we completed the 3xSMR analysis using the brain ESMR results from the previously performed analyses described above.

We used the same quality control applied in TWASs, i.e., GWAS variants with MAF below 1%, variants and probes within the MHC region, and variants with inconsistent alleles or MAF differences > 0.20 amongst pairs of four input datasets (GWAS, 1000 Genomes Project, eQTL and mQTL datasets) were excluded.

DNA: deoxyribonucleic acid; POAG: primary open-angle glaucoma; GWAS: Genome-wide association study; eQTL: expression quantitative trait loci; SMR: summary-data-based Mendelian randomization; mQTL: methylation quantitative trait loci; MSMR: SMR analysis of GWAS vs. mQTL; MESMR: SMR analysis of mQTL vs. eQTL; and ESMR: SMR analysis of GWAS vs. eQTL.

### 2.2. Phase Two: Functional Assessment

#### 2.2.1. Functional and Tissue Enrichment Analysis

The biological pathways through which POAG-associated loci act were examined using two approaches. First, we used DEPICT to identify the enriched gene sets, alongside the corresponding functions, and tissue types for genes in POAG-associated loci. DEPICT uses the same data as for functional predictions and gene prioritization (see phase one), for gene set enrichment analysis, and a set of 37,427 human gene expression microarrays for tissue enrichment analysis of 209 tissue/cell type annotations. For gene set enrichment analysis, we applied the Affinity Propagation Clustering algorithm (APCluster R package [42]), as previously suggested and implemented by Ligthart et al. [43]. The clustering was conducted based on pairwise correlation of gene sets. Second, the prioritized gene lists based on phase one (i.e., nearest genes, genes with nsSNPs linked to POAG loci, and significant genes from DEPICT, ESMR, MSMR, and MetaXcan analyses) were merged and used as the input to run functional enrichment analysis with the GeneMANIA algorithm [44] as previously described [23] and with the Ingenuity Pathway Analysis (IPA; QIAGEN) software (Figure 1). IPA core-analysis yields the relationships, canonical pathways, diseases, and functions most relevant to the uploaded set of prioritized genes. In addition, we also performed a sensitivity functional enrichment analysis in GeneMANIA, using a subset of genes that showed two or more lines of evidence out of all five approaches that we used (see methods). Furthermore, we used the OTDB described by Wagner et al. [45] to assess whether prioritized genes are overrepresented in ocular tissues. The database contains microarray gene expression values of >20,000 genes in ten human ocular tissues (choroid, ciliary body, cornea, iris, lens, optic nerve, optic nerve head, retina, sclera, and trabecular meshwork [TM]). We performed right-tailed Fisher’s exact tests [46] to evaluate the enrichment of prioritized genes amongst the genes with gene expression values in the top 25% [47].

#### 2.2.2. Genetic Correlation of POAG with Other Traits

We applied the (bivariate) LD score regression method [48] to estimate the genetic correlation of POAG with 597 UK Biobank traits with available GWAS summary statistics. POAG GWAS summary statistics data (~7.7 million SNPs) were uploaded to LDHub, an online tool dedicated to estimating the genetic correlation of traits of interest [49].

## 3. Results

### 3.1. Phase One: Gene Prioritization

#### 3.1.1. In Silico Sequencing

In silico sequencing of 50 independent lead SNPs (gSNPs) from Choquet et al. [5] returned 3250 and 1493 SNPs that are, respectively, in moderate (*r*^2^ > 0.50) and high (*r*^2^ > 0.80) LD (Appendix A). One hundred and ten of the aforementioned 1493 SNPs were in complete LD (*r*^2^ = 1), indicating that these linked SNPs represent the same association signal. Annotation of these 3250 linked SNPs using the ANNOVAR software [26] detected nine nsSNPs. Two of the nine nsSNPs (rs3753841 and rs2274224) were in perfect LD (*r*^2^ = 1) with the corresponding linked gSNPs (rs993471 and rs3891783, respectively; Appendix A). These nine nsSNPs were located in seven genes, of which one (ACP2) was novel for POAG and four (ACP2, SH2B3, SIX6, and C14orf39) did not overlap with the nearest gene list of the 50 gSNPs (Table 2).

#### 3.1.2. Co-Regulation Analyses Using DEPICT

DEPICT prioritized 86 co-regulated genes, out of 119 genes in POAG loci (*r^2^* > 0.5), suggesting their related roles in the etiology of POAG (Appendix A). Of the 86 prioritized genes, 41 were novel for POAG (Table 2). In addition, only 14 out of 86 (16%) overlapped with the nearest genes. Among these, five genes (*FBXO32*, *PLCE1*, *ARHGEF12*, *LPP*, and *BICC1*; *p* ≤ 8.75 × 10^−7^, FDR < 0.01) showed higher evidence of functional involvement in POAG pathogenesis (Appendix A). Interestingly, DEPICT also prioritized three of the seven genes with linked nsSNPs (*COL11A1*, *CAV2*, and *PLCE1*), all at *p* ≤ 1.49 × 10^−9^ (Table 2).

#### 3.1.3. MetaXcan

After the quality control steps, 11,397 variant sets were used to impute gene expression levels using the Depression Gene Network (DGN) transcriptome model. From the total of 14 significant genes, whose predicted gene expressions were associated with POAG, two genes with prediction performance *p* ≥ 3.57 × 10^−3^ (0.05/14) were filtered out and 12 significant genes remained, of which eight showed acceptable evidence of co-localization signals (Table 3). Three colocalized genes identified in MetaXcan analysis (*NR1H3*, *LTBP3*, and *EHBP1L1*) were also significant in SMR analysis (see below) of blood eQTL (Table 2).

#### 3.1.4. SMR Based on Gene-Expression Data

After applying the quality control criteria, 15,352 genes were retained for the SMR analysis. Using blood gene expression profile as a source, our analysis yielded 18 genes previously implicated/associated with glaucoma. Thirteen out of these 18 have no significant evidence of linkage confounding (based on heterogeneity in dependent instruments [HEIDI] *p* ≥ 2.78 × 10^−3^; Table 4). Three (*BICC1*, *LTBP3,* and *ABCA1)* out of these 13 significant genes were also identified by at least two additional prioritization methods. Nine out of the 13 identified genes were novel for POAG (Table 2).

Similarly, in the brain eQTL, 7361 genes were eligible for SMR analysis. Our analysis returned 10 genes whose expression profile was significantly associated with glaucoma; eight of which have no significant evidence of linkage confounding (based on heterogeneity test, HEIDI *p* ≥ 5 × 10^−3^) (Table 4). Two of these eight genes, *TXNRD2* and *CDKN2B-AS1*, overlapped with nearest genes and one (CDKN2B) with genes prioritized in the DEPICT analysis. Interestingly, the significant association of three HEIDI-passed genes (*LRRC37A2*, *LRRC37A4P*, and *RP11-707O23.5*) was also confirmed by SMR analysis of blood eQTL.

#### 3.1.5. SMR Based on Methylation Data and 3xSMR

Methylation QTL (mQTL)-based SMR analysis (MSMR) of blood and brain data yielded, respectively, 14 (SMR *p* < 5.60 × 10^−7^) and 16 (SMR *p* < 5.39 × 10^−7^) methylation sites significantly associated with glaucoma, with no evidence of LD contamination (based on heterogeneity test; HEIDI *p* ≥ 1.28 × 10^−3^ and HEIDI *p* ≥ 1.39 × 10^−3^, respectively; Table 5). Collectively, our SMR analysis of POAG GWAS and mQTL of blood and brain identified 27 novel CpG sites in 15 genes. Of these, two genes are new (*AFAP1-AS1* and *TBKBP1*), i.e., were not identified by the previous prioritization methods. Amongst significant MSMR genes, *AFAP1-AS1* and *NR1H3* are novel for POAG (Table 2). The integrative analysis of mQTL and eQTL data in the blood (MESMR) resulted in 32,420 DNA methylation (DNAm) sites significantly associated (SMRp < 1.85 × 10^−8^; regarding Bonferroni correction based on 2,697,257 tests) with expression levels of 10,680 genes not rejected by the HEIDI test (*p* ≥ 3.26 × 10^−7^). These results were used to link glaucoma-associated gene expressions to glaucoma-associated methylation levels, and identified the BICC1 gene with its genetic regulation in glaucoma to be explained by a likely causal chain (DNA→Methylation→Expression→Glaucoma). The integrative analysis of mQTL and eQTL data in the brain (MESMR) resulted in 10,685 DNAm sites significantly associated (SMRp < 1.44 × 10^−8^; regarding Bonferroni correction based on 3,482,629 tests) with expression levels of 3305 genes not rejected by the HEIDI test (*p* ≥ 2.16 × 10^−6^). We did not detect a likely causal chain from DNA to glaucoma through methylation and expression in the brain.

#### 3.1.6. Integration of Results (Phase One)

Taken all these results together, *AFAP1* and *BICC1* were simultaneously highlighted in five of the six approaches and 34 of the 142 prioritized genes were supported by at least two lines of evidence (Table 2), suggesting that these genes are involved in glaucoma development by multiple mechanisms including coding, transcription, and manipulation of targeted gene expression. On the other hand, less than half of the nearest genes (*n* = 23) overlapped with prioritized genes identified in the pipeline (Table 2). That is, 92 likely causal genes identified by different prioritization approaches were not the genes nearest to the lead gSNPs.

### 3.2. Phase Two: Functional Assessment

#### 3.2.1. Functional and Tissue Enrichment Analysis

We describe in this section the results of different functional and tissue enrichment analyses. The former include results from GeneMANIA Gene Ontology (GO) enrichment analysis, ingenuity pathways analysis (IPA) functional enrichment analysis of its manually curated biological pathways, DEPICT gene set enrichment (GSE) analysis, and affinity propagation clustering (APC) of its enriched gene sets. The latter, i.e., tissue prioritization, is based on DEPICT tissue enrichment analysis with its expression data from 209 cell types/tissues, and evaluating gene expression of the prioritized genes in 10 ocular tissues. The results of these analyses are described in order below.

GeneMANIA functional enrichment analysis revealed 246 significantly enriched gene ontology (GO) terms for 142 prioritized genes (Appendix A). Sensitivity analysis in the 34 genes with two or more lines of evidence yielded 174 significantly enriched GO terms (Appendix A). Along with several closely related pathways, most of the most significant GO terms were related to the extracellular matrix (ECM) (GO:0031012, *q* = 6.95 × 10^−56^), transforming growth factor-β (GO:0007179, q = 4.20 × 10^−11^), cardiovascular traits (e.g., blood vessel development, GO:0001568, *q*-value=2.40 × 10^−7^), heart development (GO:0007507, *q* = 2.52 × 10^−6^), Wnt signaling (GO:0016055, q = 2.52 × 10^−6^), retinoic acid receptor binding (GO:0042974, *q* = 1.18 × 10^−4^), and eye development (GO:001654, *q* = 3.09 × 10^−3^; Appendix A). Several of these pathways, including heart development, regulation of protein phosphorylation, embryo development, and Wnt-protein signaling, were in line with the most prioritized terms from DEPICT. Similarly, IPA identified 64 canonical signaling pathways (Appendix A). Based on the high percentage of focus molecules in our datasets, RAR activation (*p* = 1.32 × 10^−5^), leptin signaling (*p* = 7.94 × 10^−4^), and aryl hydrocarbon receptor signaling (*p* = 1.09 × 10^−3^) were the most strongly enriched canonical signaling pathways constructed in IPA.

Using the full set of POAG GWAS-summary statistics [5], DEPICT’s functional enrichment (FE) analysis identified 269 biological pathways enriched by glaucoma-associated loci (FDR < 0.05; Appendix A). These pathways were based on five annotation categories (see Figure 3), which include Gene Ontology (GO), Kyoto Encyclopedia of Gene and Genomes (KE), REACTOME (RE), Mouse Phenotypes (MP), and Protein-Protein Interactions (PI). Cardiovascular-related terms, e.g., abnormal aorta morphology from the MP pathway resource, and Wnt and TGF-β signaling terms from the KE pathway resource, were significantly enriched at FDR < 0.05 (Figure 3), suggesting that dysregulation in any of these pathways may contribute to POAG development. For the sake of briefness, we only visualized the top five gene sets per each annotation category (Figure 3). Furthermore, affinity propagation clustering of enriched gene sets yielded 37 gene set clusters at FDR < 0.05, including artery morphology, vasculature development and cell motility regulation. The list of all cluster centers (nodes) and pairwise Pearson correlation (edges) between the nodes is summarized in Figure 4. In the DEPICT TE (tissue/cell type) enrichment analysis, 20 tissues were prioritized at FDR < 0.05 including tissues from the urogenital system and more specifically female reproductive organs, aortic and heart valves, and arteries (Appendix A). Based on the assumed relevance for POAG, here we only show results for sense organ tissues (SO), exclusively those from the eye, as well as from the top ten nervous system tissues (Figure 3).

Our ocular tissue database (OTDB) investigations using Fisher’s exact test showed a statistically significant overrepresentation of prioritized genes in four (sclera, TM, ciliary body, and choroid) of the 10 ocular tissues (*p*-value < 0.005; i.e., 0.05/10) (Figure 5). More specifically, prioritized genes were overrepresented in the sclera, TM, and ciliary body by >1.5 fold.

#### 3.2.2. In Silico Pleiotropy Look-Up and Genetic Correlation

In silico pleiotropy analysis using GWAS catalog with European subpopulation yielded 139 linked SNPs (LD, *r^2^* > 0.80), that were previously associated with complex disease types including 41 with glaucoma itself. A complete list of all the traits identified by the in silico pleiotropic look-up along with the linked SNPs (*r*^2^ > 0.50) and their nearest genes is summarized in Appendix A. In Figure 6, we more specifically present the number of shared highly linked SNPs (*r*^2^ > 0.80) between glaucoma and six phenotypic categories that have been hypothesized in the literature to have a relationship with glaucoma, i.e., traits and diseases related to the eye (e.g., intraocular pressure) [50], anthropometry (e.g., body mass index) [22], blood pressure [19], lipids and type 2 diabetes [22], neurodegenerative disorder (e.g., Alzheimer’s disease) [21], and psychiatry (e.g., depression) [51]. Compared to other traits and diseases, POAG showed the highest genetic overlap with IOP and vertical cup-to-disc ratio, both of which are glaucoma endophenotypes with considerable heritability [2]. This can be considered as a proof of concept and internal validation of the pipeline.

We also visualized genetic correlations (rg) of POAG with traits from six phenotypic categories (Figure 7). POAG showed significant correlations with cardiometabolic disease (hypertension, rg = 0.08, *p* = 0.011), blood pressure (systolic blood pressure, rg = 0.06, *p* = 0.039; high blood pressure, rg = 0.08, *p* = 0.012) and eye diseases (diabetes-related eye disease, rg = 0.20, *p* = 0.025, other eye problems, rg = 0.44, *p* = 4.44 × 10^−9^, and senile cataract, rg = 0.27, *p* = 0.022; Figure 7). No significant genetic correlations were observed between glaucoma and Alzheimer’s disease or depression. Except for other eye problems, correlations were no longer significant after correcting for multiple testing of 597 UK Biobank traits.

#### 3.2.3. Integration of Results (Phase Two)

Some signaling pathways, e.g., developmental (Wnt/β-catenin signaling, *p* = 2.82 × 10^−3^), retinoic acid receptor activation (*p* = 1.32 × 10^−5^), and cardiac-related (Cardiac Hypertrophy Signaling, *p* = 1.10 × 10^−2^), were in common with biological processes enriched in DEPICT and GeneMANIA. In line with functional enrichment results, tissue investigations also highlighted the eye, as well as cardiovascular tissues, to be the appropriate contexts of POAG genes. In completion, pleiotropy and genetic correlation analyses also revealed shared genetic loci between glaucoma and IOP, cup-to-disc ratio, as well as blood pressure.

## 4. Discussion

We aimed to prioritize the most likely causal genes and identify the underlying biological processes involved in POAG through post-GWAS bioinformatics analyses. Our systematic post-GWAS approach spotted 142 genes as the most likely causal and/or relevant genes for POAG, 64 of which were novel. Among the prioritized genes were seven genes with nsSNPs linked to POAG genomic loci (LD *r*^2^ > 0.5), and 34 (23.9%) of the prioritized genes were supported by at least two lines of evidence. With considerable overlap, DEPICT and GeneMANIA functional enrichment analysis revealed 269 and 246 signaling pathways associated with glaucoma, respectively. ECM was the most significant and repeatedly implicated pathway in GeneMANIA. Along with several closely related pathways, TGF-β signaling, blood vessel development, heart development, and retinoic acid receptor signaling were also significantly overrepresented. Furthermore, tissues from the female reproductive as well as the cardiovascular system were significantly enriched for POAG genes. POAG-prioritized genes were overrepresented in four ocular tissues: sclera, ciliary body, TM, and choroid.

### 4.1. Highlights of Separate Analyses

Six of the 64 novel POAG genes (*NR1H3*, *ACP2*, *EHBP1L1*, *LRRC37A2*, *LRRC37A4P*, and *RP11-707O23.5*) presented two or more lines of evidence (Table 2). NR1H3 is one of the top genes associated with IOP [52], a prominent risk factor for POAG, whereas ACP2 is found to be overexpressed in the cerebellum and brain stem in neuronal ceroid lipofuscinoses (CLN3) mice [53]. CLN3 disease is an inherited disorder that affects the nervous system, and children with this disorder are characterized by progressive neurological degeneration and vision loss. Using data derived from GWAS of large consortia, previous reports showed the association of *EHBP1L1* with myopia and BP (diastolic BP and mean arterial pressure) [54,55]. Both myopia [56] and BP [19] are risk factors of glaucoma. Further studies are required to uncover how these novel genes are relevant to POAG.

Of the nine linked nsSNPs identified, two (rs3753841 and rs2274224) candidate variants were perfectly linked (LD *r*^2^ = 1) to POAG lead gSNPs, and rs3753841 mapping to COL11A1 gene was predicted to be ‘deleterious’ and ‘possibly damaging’ by SIFT [27] and Polyphen [29], respectively. Indeed, SNP rs3753841 has been reported previously to be associated with glaucoma [57,58]. This result provides evidence that rs3753841 alters an amino acid sequence in the collagen α chain precursor protein, a major component of the ECM, and contributes to susceptibility of glaucoma. This may be explained through the contribution of the modified collagen to IOP induction by generating outflow resistance in aqueous humor [59].

Furthermore, nsSNP rs8940 has previously been annotated in the coding region of *CAV2* in Australian and Swedish glaucoma patients, but its association with POAG was not significant after adjusting for the effects of a correlated (LD *r*^2^ = 0.63) SNP rs4236601 [60]. *CAV2* codes for caveolin protein family members, which is a specialized plasma membrane raft forming flask-shaped invaginations. It is involved in cell proliferation, transcellular transport, membrane lipid homeostasis, mechanotransduction, and signal transduction [61]. Caveolin 1 and caveolin 2 inhibit endothelial nitric oxide synthase enzyme activity within the caveolae, and alteration of this pathway has been associated with abnormal nitric oxide generation and TM function [62]. Caveolae are available in different retinal cell types including retinal vascular, retinal pigment epithelium, and Müller glia cells [61]. Cav-1, the principal protein of caveolae, modulates neuroprotective responses, and ablation of Cav-1 in mice and zebrafish was associated with defects in retinal pigment epithelium differentiation and STAT3 activation in the retina [63,64].

Of the 50 nearest genes identified in GWAS, only 23 (46%) genes were detected in the gene prioritization analyses, suggesting that not all nearest genes have a functional impact on protein coding, gene expression, or the regulation of transcription. This highlights the limited mapping resolution of GWAS results due to the complicated linkage disequilibrium structure of the genome, i.e., GWAS lead SNPs are not always the causal functional variants. This has been confirmed by a previous study which reported about 80% of the common GWAS variants are within 33.5 Kbp of the underlying causal variants [65]. Furthermore, our SMR analysis using methylation data also detected significant loci, strengthening the hypothesis that while SNPs located in the coding region may act by altering amino acid sequence and protein functions, genetic variants in the non-coding region may cause diseases through other mechanisms, including methylation and regulation of gene expression [66].

Whilst the role of epigenetics is well studied in mental disorders [67] and cancer [68], exploration of the role of DNA methylation in eye diseases is limited [69]. One study reported the association of a CpG site at *CDKN2B* in normal-tension glaucoma [70], another study found CpG sites at *SKI* and *GTF2H4* in the retinal pigment epithelium of age-related macular degeneration patients [71]. Our combined SMR analysis of POAG GWAS data and mQTL of blood and brain data identified 27 novel DNA methylation sites for POAG in 15 genes, highlighting the role of epigenetics in gene expression and glaucoma. Moreover, our 3xSMR analysis, which was used for linking glaucoma-associated gene expression to glaucoma-associated methylation levels, enabled us to find two methylation sites (cg05938607 and cg12342675) at the BICC1 gene, confirming its likely causal role in methylation mediated genetic regulation in glaucoma (DNA→methylation→expression→glaucoma; Figure 2).

### 4.2. Post-GWAS Analyses Yielding Most Reliable Data

Three genetic loci (*AFAP1*, *BICC1*, and *ABCA1*) were identified in four or more approaches in our pipeline, confirming earlier evidence that these genes have a likely role in glaucoma pathogenesis. Actin filament associated protein 1 (*AFAP1*), also known as *AFAP110* or *AFAP-110*, encodes the nonreceptor tyrosine kinase (Src) binding partner protein and affects actin filament organization in response to cellular signals [72]. Activation of Src signaling, in turn, has been implicated in the attenuation of ECM degradation via the inhibition of plasminogen activator expression, and the application of dasatinib, a potent Src signaling inhibitor, in rats improved the TGF-β2-induced adhesive and contractile characteristics of the TM and also attenuated ECM deposition [73]. BICC1, Bicaudal-C (BicC) family RNA binding protein 1, is involved in gene expression during embryonic development and is a negative regulator of the Wnt signaling pathway [74]. Previous studies reported the association of BICC1 with POAG [5,75], major depressive disorder [76], and high myopia [77]. The GTEx, CAGE, and FANTOM5 projects showed high average RNA expression levels in the eye [78]. BICC1 and AFAP1 were the two most highly prioritized genes; we further searched DEPICT gene set enrichment results in order to find the most likely relevant pathways through which these two genes act. Our co-regulation analyses predicted the highest functional similarity of *BICC1* with an ECM-related cluster of gene sets. Within the cluster, the strongest correlation of *BICC1* points towards abnormal tendon morphology, with corneal thinning also among the gene sets in the cluster (Figure 8A). In addition to cell motility, our analysis predicts *AFAP1’s* potential involvement in vasculature development as well as neuron differentiation (Figure 8B).

ATP-binding cassette transporter A1 (*ABCA1*) protein is a cholesterol and phospholipid transporter across the cell membrane, and mutation of this gene is associated with cancer and lipoprotein metabolism abnormality [79,80]. Although the association of *AFAP1* and *ABCA1* genes with cancer has been widely studied [81,82], there is a paucity of data showing the underlying mechanism in glaucoma. Cui et al. used reverse transcription polymerase chain reaction and immunolabeling approaches to examine the expression of AFAP1, ABCA1, and GMDS in human ocular tissues. Both *AFAP1* and *ABCA1* showed significant gene expression in the retina, retinal ganglion, optic nerve, and TM cells [83]. Our TWAS analyses showed that in contrast to the potential protective effect of increased *AFAP1* expression, *ABCA1* expression level is positively correlated with the risk of POAG. In line with this result, there is supporting evidence that *ABCA1* is involved in the apoptosis of retinal ganglion cells in rat glaucoma models [84].

### 4.3. Functional Assessments

POAG is a multifactorial disease with multiple pathways involving different tissues. In normal human development, complex signaling pathways control cells’ proliferation, differentiation and fate, motility, and death; thus hypo/hyperactivation of either of these pathways may lead to the development of chronic diseases including POAG [85]. Functional enrichment analysis showed that the 142 prioritized causal genes act through several pathways, but most importantly, candidate genes were overrepresented in ECM, TGF-β signaling, blood vessel development, heart development, angiogenesis, and the retinoic acid receptor signaling pathway. Overrepresentation of the ECM and TGF-β pathways is in line with previous in vitro studies which reported TGF-β-induced morphological changes in the ocular structures, and possibly the development of POAG, using human TM cells as well as optic nerve astrocytes [86].

The ECM is a group of more than 300 complex and dynamic proteins regulating many biological processes and structures. The composition of ECM is unique in each organ, and it is continuously remodeled to regulate tissue homeostasis [87]. Previous studies have suggested that the cross-linking and deposition of ECM proteins in TM are responsible for aqueous humor outflow resistance and IOP elevation in glaucoma patients [86]. Additionally, glaucoma patients have elevated levels of TGF-β in their aqueous humor, and high TGF-β has been shown to increase the synthesis of ECM in human TM cells [86,88]. Few studies have described the mechanisms and cascade of events in how ECM leads to TM dysfunction and increased IOP. Prior experimental studies reported that the induction of ECM cross-linking LOX genes by three TGF-β isoform proteins change ECM stiffness in human glaucomatous TM cells. The authors inferred that TGF-β-mediated overexpression of LOX activity is partially responsible for the increased aqueous humor outflow resistance and IOP elevation [89].

TGF-β is a polypeptide whose signaling has a pleiotropic effect in several cell functions depending on the cellular context. For example, TGF-β receptor signaling, mediated via canonical SMAD pathways, controls the expression of hundreds of genes, while via non-canonical pathways, it regulates cell polarity and microRNA maturation [90,91]. Although TGF-β in normal tissues has pleiotropic roles in multiple biological processes and does not lead to fibrosis, it has a major impact on fibrosis and scarring following glaucoma surgery [92]. Of the three TGF-β isoforms (TGF-β1, TGF-β2, and TGF-β3), increased levels of TGF-β2 in the aqueous humor was associated with fibrosis of TM in POAG patients [93]. In addition, a recent study reported that the TGF-β-induced increase in collagen expression by TM cells is linked with phosphorylation of PTEN, and the manipulation of PTEN activity has been suggested as having therapeutic potential to prevent fibrosis of TM in POAG patients [94]. Other studies demonstrated that TGF-β2 increases expression of PAI-1 and prevents the activation of matrix metalloproteinases (enzymes responsible for the degradation of most ECM proteins) via both tissue-type (tPA) and urokinase-type urokinase (uPA) plasmin activators, thereby increasing the cross-linking of ECM components in TM cells via transglutaminase [95,96]. Another in vitro study demonstrated a cascade of events initiated by TGF-β2 could lead to ECM deposition. TGF-β2 injected into human TM cultured cells significantly increased fibronectin (ECM glycoprotein) levels. In non-glaucomatous eyes, TM cells secrete BMP-4 and antagonize the TGF-β2 pathway. In contrast, in POAG eyes, Gremlin (glycoprotein) is overexpressed in the TM cells, and this inhibits the antagonistic effect of BMP-4 on TGF-β2, thereby leading to increased ECM crosslinking and elevated IOP [97].

Overrepresentation of prioritized genes in cardiovascular-related pathways is consistent with the fact that abnormal blood vessel growth, through the activation of vascular endothelial growth factor (VEGF), has been implicated in neovascular glaucoma, age-related macular degeneration, and diabetic retinopathy [98]. Similarly, insufficient blood supply to the optic nerve head, either due to elevated IOP or low BP, is a prominent risk factor put forward to explain the development and progression of glaucoma through epidemiological studies [99]. Wnt signaling, combined with several signaling pathways, controls crucial tissues and organs during embryonic development [100]. Accordingly, a recent review showed that Wnt signaling has a vital role in angiogenesis and vascular morphogenesis in eye development, and increased activation of this signal was implicated in diabetic retinopathy, wet age-related macular degeneration, corneal neovascularization, and retinopathy of prematurity [101].

Our functional analysis implied the significance of retinoic acid (RA) and its signaling pathways in glaucoma pathogenesis. RA, a derivative of vitamin A, is essential for regulating genetic transcription that controls a wide range of biological processes including development and homeostasis, as well as cellular apoptosis and survival [102]. The identification of this pathway may strengthen the previous evidence that reported the involvement of *CYP26A1* (which was included among the list of prioritized genes) and *CYP26C1* in the regulation of RA metabolism, eye development, and maturation of visual function [103]. Moreover, RAs and their receptors (RARs and RXRs) have an important role in eye development and physiology [104], and in a recent study, Prat et al. demonstrated that myocilin (a pathogenic genetic biomarker for glaucoma) expression is regulated by RA via the RARE-DR2 promotor [105]. Furthermore, intraocular injection of RA receptor agonists significantly reduced the number of injured axons of the retinal ganglion cells in frogs, while the injection of antagonists and also the inhibition of RA synthesis with disulfiram had the opposite effect [106]. The role of RA signaling in the pathogenesis of POAG via myocilin transcription, or through its direct effect on retinal ganglion cells, may provide clues for new therapeutic approaches in glaucoma. Our findings highlight the urogenital system and, more specifically, female reproductive organs as tissues in which POAG genes are highly expressed. This is in line with the previous evidence of a protective role for female hormones in glaucoma [107,108,109]. Aortic and heart valves, which were among the most enriched tissues, support the results of our functional enrichment analysis regarding vasculature development and artery morphology, as well as previous studies [10]. These observations may also partly explain the shared genetics that we observed for glaucoma and blood pressure. Additional research is required to elucidate the relevance of digestive tract systems and to determine if this is only explained by the potential role of the gut microbiota in glaucoma progression, through the gut–retina axis [110]. Furthermore, the expression of POAG genes in sclera and choroid is in agreement with previous studies that showed an association between ocular rigidity (structural stiffness of the ocular tissues, including sclera, choroid, and cornea) and glaucoma [111,112]. For example, the biomechanical response of the optic nerve head to IOP stress is determined by the mechanical properties of the sclera, i.e., the sclera plays an important role in transmitting forces created by changes in IOP to the optic nerve head, thereby linking the pressure–glaucoma damage pathway [113,114].

In silico pleiotropy look-ups in the GWAS catalog confirmed the already-known genetic overlap between POAG and its related endophenotypes. For example, sixty-five linked glaucoma SNPs showed a pleiotropic effect on IOP, which is a modifiable risk factor for glaucoma (Figure 6). In addition, several glaucoma-linked SNPs were associated with optic disc parameters, including optic cup area, optic disc area, and vertical cup-to-disc ratio, a clinical metric of glaucoma [115].

Previous epidemiological studies have also suggested a close relationship between glaucoma and the two most common neurodegenerative diseases (Alzheimer’s and Parkinson’s disease), suggesting that glaucoma is the consequence of an age-related neurodegenerative process that affects the visual system [21,116]. Additionally, clinical studies have demonstrated that extracellular amyloid β (Aβ) accumulation, hyperphosphorylated tau protein (pTau) aggregation, and glial reaction are common pathological mechanisms in both glaucoma and Alzheimer’s diseases [117,118]. However, our pleiotropic look-ups and bivariate LD score regression analysis demonstrated no significant genetic correlation, highlighting that non-genetic factors (environmental factors), or undiscovered confounders, might account for the reported relationships. Our functional enrichment analysis outcomes agreed with previous comparable studies that reported the overrepresentation of ECM, TGF-β, lipid metabolism, developmental, and cardiovascular-related pathways in glaucoma, but could not confirm the importance of inflammation-related factors (e.g., nuclear factor-κB (NFκB), and tumor necrosis factor α (TNF-α)) [10,11]. The most plausible explanation for the apparent discrepancy, regarding the inflammation-related pathway, is the difference in the gene prioritization methodology used. We included genes that may have a role through different mechanisms (coding consequence, gene expression, co-regulation, and epigenetic regulation), whereas the data used as the input for the functional enrichment analysis in Janssen et al. [11] and Danford et al. [10] were significant genes curated from previous linkage, candidate-gene, and genome-wide association studies. IPA [10,11] and ConsensuspathDB [10] were used for functional annotation, whereas in addition to IPA, we used DEPICT as well as the GeneMANIA algorithm, along with its corresponding composite networks based on about 600 million interactions, which may also partly explain the difference in the identified functional pathways.

## 5. Strengths and Limitations

Despite the very large sample sizes of our input data sets and different statistical approaches used to find likely causal genes and pathways, there are still some limitations regarding our work. More specifically, we had to use expression data from the blood and brain to increase power for TWAS analyses, because only very limited sample sizes are available for eye tissues. Inevitably, as in any other research, our work is built on the basis of the previous studies in terms of data or methods, which may have biased our findings towards previous knowledge. However, we used agnostic methods and included data from large-scale studies, offering ample opportunities to uncover new aspects of the disease, while concurrently avoiding false interpretations by using conservative significance cut-off values by adjusting for multiple hypothesis testing. We acknowledge that while we were conducting this research, 19 (*NR1H3*, *ACP2*, *EHBP1L1*, *ANGPTL2*, *CALD1*, *CLIC5*, *COL16A1*, *H1F0*, *KLF5*, *KREMEN1*, *LOC100147773*, *MAPT*, *MIR4778*, *MNAT1*, *NPEPPS*, *SLC2A12*, *THSD4*, *TRIB2*, *TRIOBP*) of the 64 novel prioritized genes were detected in two recent POAG GWAS publications [119,120].

## 6. Conclusions

We shed light on the underlying biology of glaucoma in terms of likely involved genes and pathways using the most comprehensive recent GWAS on POAG. Our prioritized genes, particularly those with multiple lines of evidence, are eligible for further in vitro and in vivo studies. Such studies, together with improved knowledge on highlighted biological pathways including ECM organization in TM, retinoic acid signaling as well as blood vessel development, may ultimately yield new and more effective therapies.

## Figures and Tables

**Figure 2 genes-13-01055-f002:**
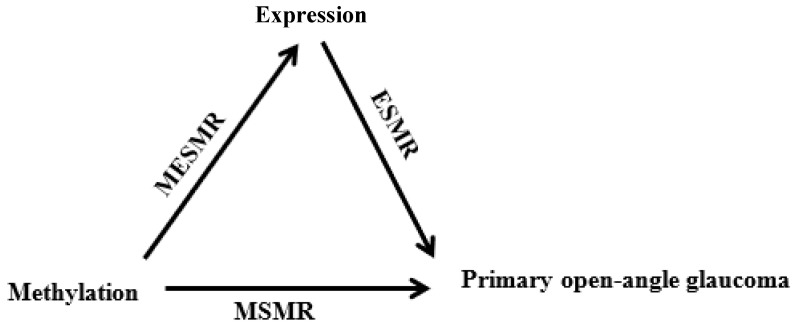
Diagrammatic representation of 3xSMR analysis showing the cascade of causal associations from DNA methylation to POAG development.

**Figure 3 genes-13-01055-f003:**
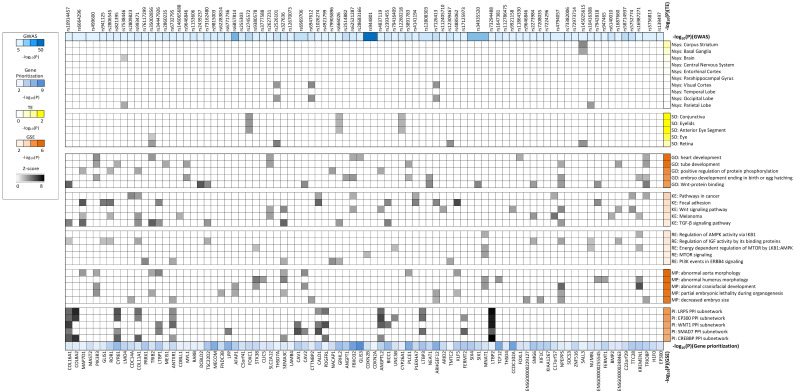
Heatmap of DEPICT results: DEPICT prioritized genes (FDR < 0.05) in the context of most significant DEPICT gene sets and likely relevant tissues are visualized. Gray-scale colors represent each gene’s contribution to gene-set enrichment (GSE) or tissue enrichment (TE) described as a Z-score (only the top ten Z-scores per gene set/tissue are shown). For each of the five annotation categories (GO, KE, RE, MP, PI), only the top five gene sets are visualized. Sidebars represent *p*-values on the logit scale.

**Figure 4 genes-13-01055-f004:**
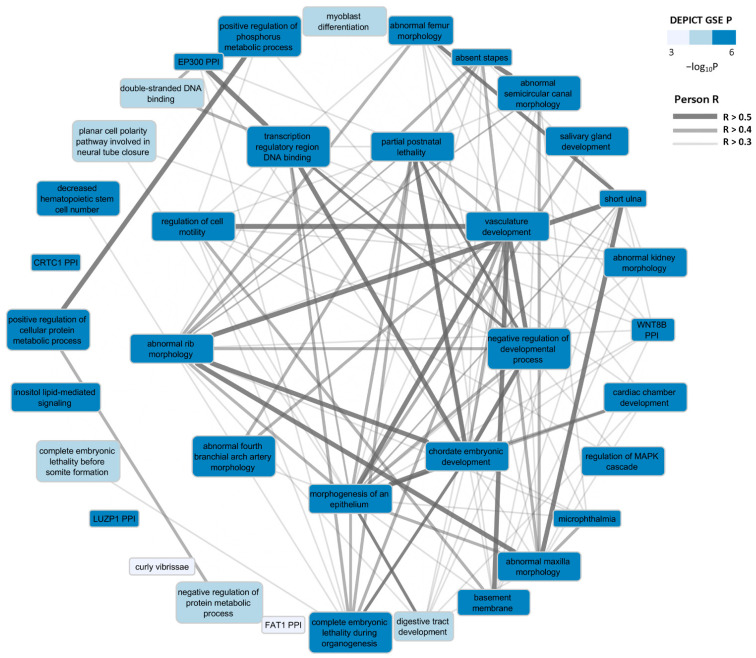
DEPICT network plot: nodes represent enriched gene sets (FDR < 0.05) and links represent pairwise Pearson correlation coefficient (r) between gene sets. Only cluster centers from affinity-propagation clustering and only links with r > 0.3 are shown. The inner layer contains pathways with centrality degree ≥ 26, i.e., the 4th quartile of all gene sets, meaning the largest number of connections which imply their importance in network survival.

**Figure 5 genes-13-01055-f005:**
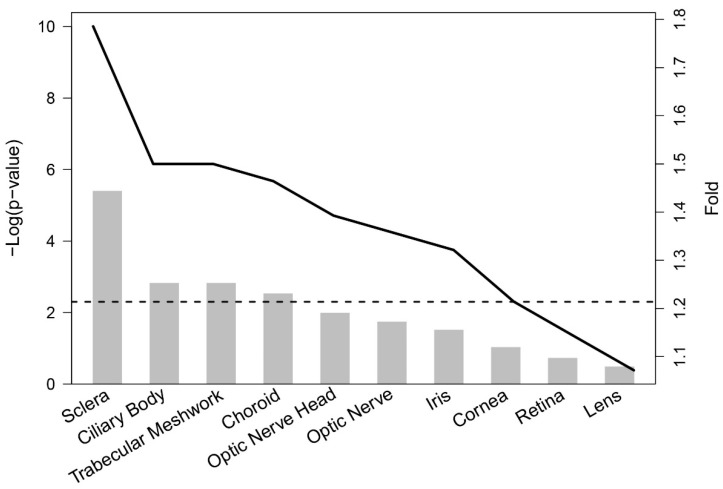
Bar graph showing enrichment in ten ocular tissues of 112 available prioritized genes (out of the total of 142) amongst the genes in the ocular tissue database with gene expression values in the top 25%. Bars show the significance of overrepresentation (y-axis left), dashed horizontal line running through the bars is the threshold for *p*-value < 0.005 (adjusted for multiple testing), and the solid black line shows the fold of overrepresentation (y-axis right).

**Figure 6 genes-13-01055-f006:**
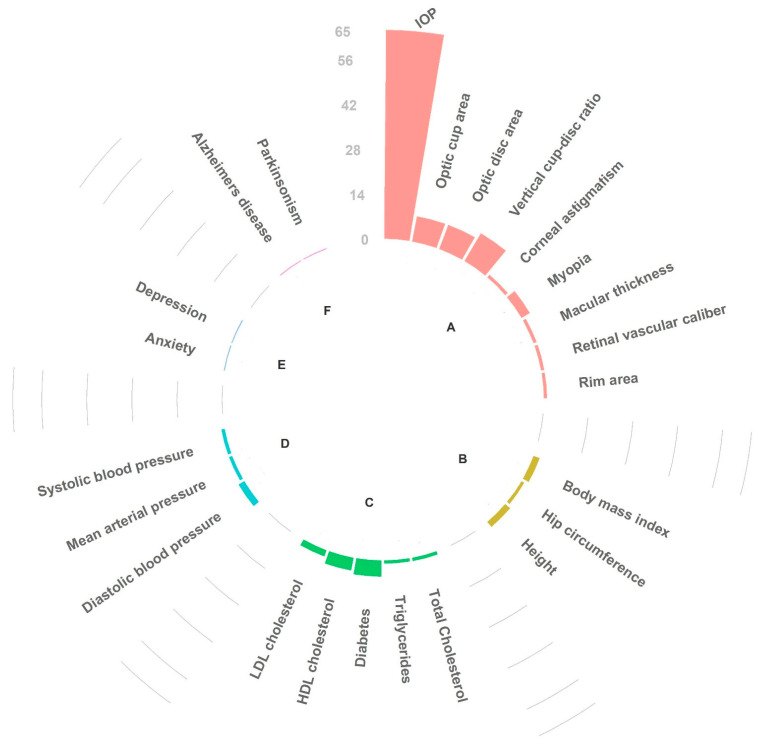
Circular barplot: each bar shows the number of linked SNPs (LD *r*^2^ > 0.80) having pleiotropic effects between POAG-associated loci and six phenotypic categories. A: optic parameters and other eye problems; B: anthropometry-related traits; C: lipids and type 2 diabetes; D: blood pressure; E: psychiatric disorders; F: neurodegenerative disorders.

**Figure 7 genes-13-01055-f007:**
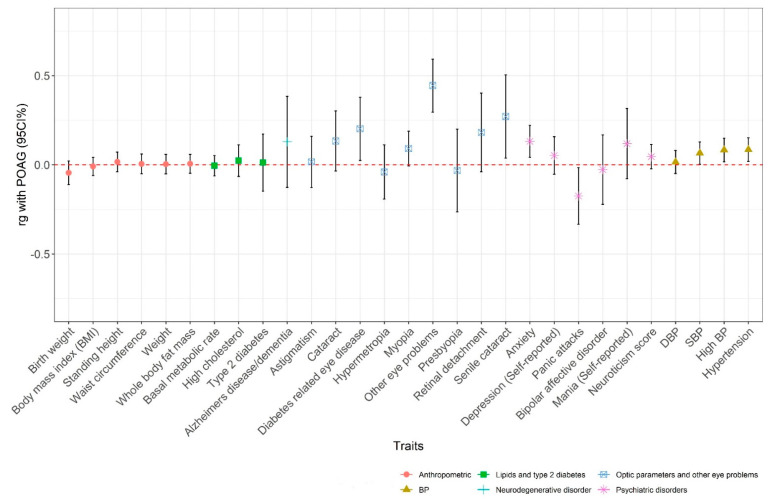
Dot and whisker plot: the genetic correlations and 95% confidence intervals between POAG and six phenotypic categories were calculated using the LD score regression method.

**Figure 8 genes-13-01055-f008:**
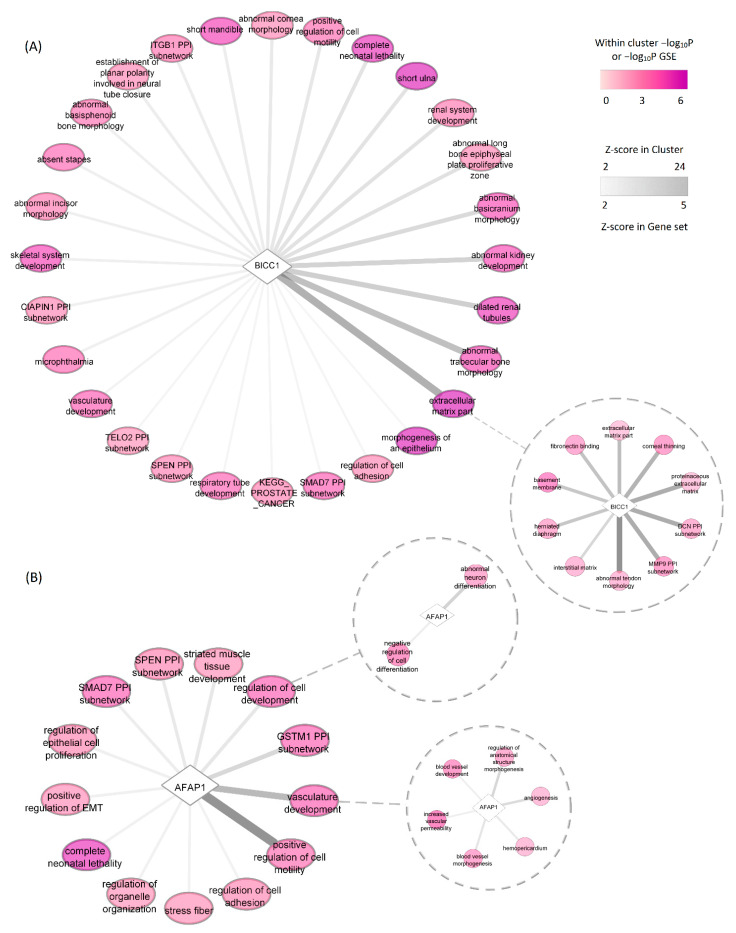
Correlation of *BICC1* (**A**) *AFAP1* (**B**) with DEPICT reconstituted gene sets. Ovals represent gene set clusters. Edges represent correlation Z-score which for gene set clusters is the sum of correlation with their individual gene sets. Detailed correlation of genes with individual gene sets (circles) within clusters is shown as subnetworks.

**Table 1 genes-13-01055-t001:** Lead POAG gSNPs as reported by Choquet et al. [5].

Ser. No.	SNP	Chr	Position	A1/A2	GWAS *p*-Value	OR (95% CI)	Nearest Gene
1	**rs41461152**	1	88227120	T/G	4.1 × 10^−9^	1.09 (1.06–1.12)	*LMO4*/*PKN2-AS1*
2	**rs993471**	1	103385373	G/A	2.0 × 10^−8^	1.08 (1.05–1.11)	*COL11A1*
3	**rs2814471**	1	165739598	C/T	2.0 × 10^−62^	1.37 (1.32–1.42)	*TMCO1*
4	rs7524755	1	165694897	T/C	8.3 × 10^−41^	1.35 (1.27–1.44)	*TMCO1*
5	**rs1192415**	1	91611540	G/A	2.4 × 10^−5^	1.12 (1.06–1.18)	*CDC7*–*TGFBR3*
6	rs4656461	1	165717968	G/A	1.2 × 10^−19^	1.33 (1.25–1.41)	*TMCO1*
7	**rs2627761**	2	55933014	C/T	6.1 × 10^−10^	1.11 (1.07–1.15)	*PNPT1*
8	**rs2860235**	2	66567896	T/C	3.9 × 10^−8^	0.93 (0.90–0.95)	*MEIS1*
9	**rs3789134**	2	111680155	T/C	1.9 × 10^−9^	1.09 (1.06–1.12)	*ACOXL*
10	**rs6434068**	2	153357541	G/C	1.9 × 10^−11^	0.91 (0.89–0.94)	*FMNL2*
11	**rs56335522**	2	213758234	G/C	1.7 × 10^−13^	1.19 (1.13–1.24)	*IKZF2*
12	**rs34339006**	2	234271522	C/T	1.7 × 10^−8^	0.93 (0.90–0.95)	*DGKD*
13	rs56117902	2	153304730	A/C	7.3 × 10^−5^	0.88 (0.84–0.92)	*FMNL2*
14	**rs1153606**	3	25574186	A/G	3.4 × 10^−8^	1.10 (1.06–1.14)	*RARB*
15	**rs9883252**	3	85138818	T/C	6.1 × 10^−14^	1.11 (1.08–1.14)	*CADM2*
16	**rs73162480**	3	150063454	G/T	1.8 × 10^−8^	1.15 (1.09–1.20)	*TSC22D2*
17	**rs9853115**	3	186131600	T/A	1.9 × 10^−13^	1.10 (1.08–1.13)	*DGKG*/*TBCCD1*
18	**rs6771736**	3	188066437	C/G	4.9 × 10^−8^	0.92 (0.89–0.95)	*LPP*
19	rs34201102	3	85137499	A/G	2.1 × 10^−5^	1.11 (1.08–1.15)	*CADM2*
20	**rs6857814**	4	7916540	A/G	5.8 × 10^−27^	0.86 (0.84–0.89)	*AFAP1*
21	rs59521811	4	7909772	T/C	9.8 × 10^−17^	0.86 (0.82–0.90)	*AFAP1*
22	rs9330348	4	7883887	C/G	5.7 × 10^−10^	1.16 (1.12–1.20)	*AFAP1*
23	rs4619890	4	7851433	G/A	0.00025	1.08 (1.04–1.13)	*AFAP1*
24	**rs76325372**	5	14837332	A/C	6.7 × 10^−13^	1.12 (1.08–1.15)	*ANKH*
25	**rs255303**	5	172588027	G/T	3.5 × 10^−8^	1.08 (1.05–1.11)	*BNIP1*
26	**rs72835984**	6	642017	C/T	1.7 × 10^−15^	0.86 (0.83–0.89)	*EXOC2*
27	**rs9392348**	6	1989604	G/A	1.7 × 10^−10^	1.11 (1.08–1.15)	*GMDS*
28	**rs17752199**	6	51406848	A/G	4.7 × 10^−9^	1.14 (1.09–1.19)	*PKHD1*
29	**rs3757155**	6	136458593	C/T	9.5 × 10^−15^	1.12 (1.09–1.15)	*PDE7B*
30	**rs4709212**	6	158976277	C/A	7.8 × 10^−9^	1.09 (1.06–1.12)	*TMEM181*
31	**rs3012455**	6	170448016	A/G	9.3 × 10^−9^	1.14 (1.09–1.20)	*FAM120B*
32	rs9494457	6	136474794	T/A	7.4 × 10^−6^	1.16 (1.11–1.22)	*PDE7B*
33	rs2073006	6	637465	C/T	1.2 × 10^−6^	0.86 (0.82–0.90)	*EXOC2*
34	**rs2745572**	6	1548134	A/G	5.2 × 10^−7^	1.12 (1.07–1.18)	*FOXC1*
35	rs11969985	6	1922673	G/A	0.00017	1.13 (1.06–1.20)	*GMDS*
36	**rs2526101**	7	11677452	A/G	1.4 × 10^−10^	0.92 (0.89–0.94)	*THSD7A*
37	**rs327636**	7	80848822	A/G	7.0 × 10^−9^	1.08 (1.06–1.12)	*SEMA3C*/*CACNA2D1*
38	**rs6969706**	7	116154831	G/T	3.8 × 10^−12^	0.90 (0.88–0.93)	*CAV2*/*CAV1*
39	**rs6947612**	7	117632975	A/G	1.2 × 10^−8^	0.92 (0.90–0.95)	*CTTNBP2*/*LSM8*
40	**rs62478245**	7	151505698	C/T	3.3 × 10^−8^	0.82 (0.76–0.88)	*PRKAG2*
41	rs12699251	7	11679113	A/G	0.017	0.90 (0.87–0.93)	*THSD7A*
42	rs4236601	7	116522675	G/A	1.0 × 10^−4^	0.91 (0.87–0.95)	*CAV2*
43	**rs2514882**	8	108275781	C/T	7.3 × 10^−12^	0.87 (0.84–0.91)	*ANGPT1*
44	**rs62521287**	8	124552133	C/T	8.3 × 10^−9^	0.86 (0.81–0.90)	*FBXO32*
45	rs2514884	8	108276873	C/T	0.014	0.84 (0.80–0.88)	*ANGPT1*
46	**rs944801**	9	22051670	G/C	2.1 × 10^−49^	0.81 (0.79–0.83)	*CDKN2B-AS1*
47	**rs2472493**	9	107695848	G/A	6.1 × 10^−29^	1.16 (1.13–1.19)	*ABCA1*
48	**rs1536907**	9	129382560	A/G	1.1 × 10^−17^	0.87 (0.84–0.90)	*LMX1B*
49	rs10811645	9	22049656	G/A	1 × 10^−18^	0.77 (0.74–0.80)	*CDKN2B-AS1*
50	rs1333037	9	22040765	C/T	2.9 × 10^−29^	0.84 (0.81–0.87)	*CDKN2B-AS1*
51	rs55770306	9	129388033	C/A	0.00019	0.86 (0.83–0.90)	*LMX1B*
52	rs4977756	9	22068653	G/A	9.7 × 10^−21^	0.81 (0.77–0.84)	*CDKN2B-AS1*
53	**rs2393455**	10	60374898	C/A	7.1 × 10^−9^	1.08 (1.05–1.11)	*BICC1*
54	**rs3891783**	10	96015793	C/G	1.6 × 10^−8^	0.93 (0.90–0.95)	*PLCE1*
55	**rs10838692**	11	47345100	T/C	8.8 × 10^−10^	0.92 (0.89–0.94)	*MADD*
56	**rs12808303**	11	65225319	C/G	3.5 × 10^−10^	1.12 (1.08–1.16)	*NEAT1*
57	**rs7126413**	11	120207405	A/G	1.1 × 10^−11^	0.91 (0.89–0.94)	*ARHGEF12*
58	rs12806740	11	120203628	G/A	0.0017	0.90 (0.87–0.93)	*TMEM136*
59	rs58073046	11	120377784	A/G	4.3 × 10^−5^	0.87 (0.82–0.93)	*ARHGEF12*
60	**rs12309467**	12	84038478	C/G	1.9 × 10^−9^	1.08 (1.06–1.11)	*TMTC2*/*SLC6A15*
61	rs324794	12	83946450	G/T	6.8 × 10^−3^	0.87 (0.83–0.91)	*TMTC2*
62	**rs7137828**	12	111494996	C/T	0.0034	0.93 (0.89–0.98)	*ATXN2*
63	**rs17075855**	13	22669058	G/A	3.8 × 10^−8^	0.91 (0.88–0.94)	*LINC00540*
64	**rs34935520**	14	61091401	G/A	7.8 × 10^−24^	1.15 (1.12–1.18)	*SIX1*/*SIX6*
65	rs35155027	14	61095174	G/C	6.2 × 10^−13^	1.17 (1.12–1.23)	*SIX1*/*SIX6*
66	rs10483727	14	60606157	T/C	1.6 × 10^−12^	1.17 (1.12–1.22)	*SIX1*/*SIX6*
67	**rs1647381**	15	57086199	C/G	8.2 × 10^−9^	0.90 (0.87–0.93)	*ZNF280D*/*TCF12*
68	**rs2245899**	15	61952501	G/A	5.1 × 10^−9^	1.08 (1.05–1.11)	*VPS13C*
69	rs2593221	15	57501414	A/G	0.025	0.86 (0.82–0.90)	*TCF12*
70	**rs9913911**	17	10031183	A/G	9.1 × 10^−34^	1.19 (1.15–1.22)	*GAS7*
71	rs9897123	17	10117184	C/T	1.1 × 10^−13^	1.18 (1.13–1.23)	*GAS7*
72	**rs6140010**	20	6473123	A/G	7.6 × 10^−10^	1.09 (1.06–1.12)	*CASC20*
73	**rs58714937**	22	19856710	C/T	1.1 × 10^−13^	1.15 (1.11–1.20)	*TXNRD2*
74	**rs5752774**	22	29105610	C/T	1.8 × 10^−11^	0.91 (0.88–0.93)	*CHEK2*
75	rs35934224	22	19885122	C/T	2.8 × 10^−5^	1.15 (1.08–1.22)	*TXNRD2*

POAG: primary open-angle glaucoma; SNP: single-nucleotide polymorphism; gSNP: single-nucleotide polymorphism from genome-wide association study; OR: odds ratio; 95% CI: 95% confidence interval; A1: effect (risk) allele; A2: non-effect allele; SNPs in bold (*n* = 50): lead independent POAG gSNPs after clumping the 75 gSNPs.

**Table 2 genes-13-01055-t002:** Number of prioritized genes per pipeline and their novelty in POAG.

Entry	Gene Symbol	Top SNP	Nearest Genes (*n* = 50)	Genes with nsSNPs (*n* = 7)	DEPICT (*n* = 86)	SMR Blood (*n* = 13)	SMR Brain (*n* = 8)	MetaXcan (*n* = 8)	MSMR Blood (*n* = 7)	MSMR Brain (*n* = 12)	Common in Different Lines	Novelty *
1	*AFAP1*	rs6857814	1		1			1	1	1	5	Known
2	*BICC1*	rs2393455	1		1	1			1	1	5	Known
3	*ABCA1*	rs2472493	1			1			1	1	4	Known
4	*ARHGEF12*	rs7126413	1		1				1		3	Known
5	*CAV2*	rs6969706	1	1	1						3	Known
6	*COL11A1*	rs993471	1	1	1						3	Known
7	*EXOC2*	rs72835984	1						1	1	3	Known
8	*LTBP3*	rs12808303			1	1		1			3	Known
9	*NR1H3*	rs326222				1		1		1	3	Novel
10	*PLCE1*	rs3891783	1	1	1						3	Known
11	*ACOXL*	rs3789134	1							1	2	Known
12	*ACP2*	rs2167079		1				1			2	Novel
13	*ANGPT1*	rs2514882	1		1						2	Known
14	*C14orf39*	rs34935520		1						1	2	Known
15	*C22orf29*	rs58714937			1			1			2	Known
16	*CDKN2B*	rs72652413			1		1				2	Known
17	*CDKN2B-AS1*	rs944801	1				1				2	Known
18	*CTTNBP2*	rs6947612	1		1						2	Known
19	*DGKD*	rs34339006	1							1	2	Known
20	*EHBP1L1*	rs1346				1		1			2	Novel
21	*FBXO32*	rs62521287	1		1						2	Known
22	*GAS7*	rs9913911	1							1	2	Known
23	*LMX1B*	rs1536907	1							1	2	Known
24	*LPP*	rs6771736	1		1						2	Known
25	*LRRC37A2*	rs112560196				1	1				2	Novel
26	*LRRC37A4P*	rs112560196				1	1				2	Novel
27	*NEAT1*	rs12808303	1		1						2	Known
28	*PDE7B*	rs9494457	1						1		2	Known
29	*RARB*	rs1153606	1		1						2	Known
30	*RP11-707O23.5*	rs112073200				1	1				2	Novel
31	*SIX6*	rs34935520		1						1	2	Known
32	*THSD7A*	rs12699251	1		1						2	Known
33	*TMTC2*	rs324794	1		1						2	Known
34	*TXNRD2*	rs35934224	1				1				2	Known
35	*AC007038.1*	rs9646846			1						1	Novel
36	*AC040170.1*	rs1687660			1						1	Novel
37	*AFAP1-AS1*	rs62290601							1		1	Novel
38	*ANGPTL2*	rs4837119			1						1	Novel
39	*ANKH*	rs76325372	1								1	Known
40	*ANTXR1*	rs6732795			1						1	Known
41	*AP004608.1*	rs7942818			1						1	Novel
42	*ARID2*	rs112405710			1						1	Known
43	*ATXN2*	rs7137828	1								1	Known
44	*BMP2*	rs2206916			1						1	Known
45	*BNIP1*	rs255303	1								1	Known
46	*C17orf57*	rs4794057			1						1	Novel
47	*C17orf69*	rs450751						1			1	Novel
48	*C5orf41*	rs255303			1						1	Novel
49	*CADM2*	rs34201102	1								1	Known
50	*CALD1*	rs1026274			1						1	Novel
51	*CASC20*	rs6140010	1								1	Known
52	*CAV1*	rs2896175			1						1	Known
53	*CCDC102A*	rs9921158			1						1	Novel
54	*CDC14A*	rs2809823			1						1	Novel
55	*CDKN2A*	rs72652413			1						1	Known
56	*CHEK2*	rs5752774	1								1	Known
57	*CLIC5*	rs3777588			1						1	Novel
58	*COBLL1*	rs146065688			1						1	Novel
59	*COL16A1*	rs10914457			1						1	Novel
60	*COL8A2*	rs6664296			1						1	Known
61	*CRHR1-IT1*	rs112560196				1					1	Novel
62	*CYP26A1*	rs12260218			1						1	Known
63	*CYR61*	rs821395			1						1	Novel
64	*DCBLD2*	rs2439237			1						1	Known
65	*DND1P1*	rs113991678				1					1	Novel
66	*EP300*	rs139497			1						1	Known
67	*FERMT1*	rs947465			1						1	Novel
68	*FERMT2*	rs17125973			1						1	Known
69	*FMNL2*	rs6434068	1								1	Known
70	*FNDC3B*	rs62283814			1						1	Known
71	*FOXC1*	rs2745572			1						1	Known
72	*FOXCUT*	rs2745572	1								1	Novel
73	*FOXL1*	rs11864330			1						1	Novel
74	*GLIS1*	rs941125			1						1	Known
75	*GLIS3*	rs28683166			1						1	Known
76	*GMDS*	rs72835984	1								1	Known
77	*GRHL2*	rs666026			1						1	Novel
78	*H1F0*	rs5756813			1						1	Novel
79	*KANSL1-AS1*	rs112073200				1					1	Novel
80	*KIAA1267*	rs115690894			1						1	Novel
81	*KIF1C*	rs7208035			1						1	Novel
82	*KLF5*	rs4885062			1						1	Novel
83	*KREMEN1*	rs16987271			1						1	Novel
84	*LAMB4*	rs12670073			1						1	Novel
85	*LINC00540*	rs17075855	1								1	Known
86	*LINC01214*	rs73162480	1								1	Novel
87	*LINC01364*	rs41461152	1								1	Novel
88	*LINC02052*	rs9883252	1								1	Known
89	*LINC02349*	rs2245899	1								1	Novel
90	*LMO4*	rs7538446			1						1	Known
91	*LOC100147773*	rs2814471	1								1	Novel
92	*LOC102724511*	rs3012455	1								1	Novel
93	*LOC105369146*	rs327636	1								1	Novel
94	*LRRC37A*	rs62641967					1				1	Novel
95	*LTBP1*	rs34447926			1						1	Novel
96	*LTBP2*	rs1077662			1						1	Known
97	*MADD*	rs10838692	1								1	Known
98	*MAP7D1*	rs6664296			1						1	Novel
99	*MAPT*	rs62641967					1				1	Novel
100	*MAST2*	rs499600			1						1	Novel
101	*MECOM*	rs13062416;			1						1	Known
102	*MEIS1*	rs2860235			1						1	Known
103	*MIR4776-2*	rs56335522	1								1	Novel
104	*MIR4778*	rs2860235	1								1	Novel
105	*MNAT1*	rs34935520			1						1	Novel
106	*MVB12B*	rs10122788				1					1	Known
107	*MYL1*	rs9646846			1						1	Novel
108	*NACAP1*	rs79905896			1						1	Novel
109	*NPEPPS*	rs4794057			1						1	Novel
110	*NUMBL*	rs10416308			1						1	Novel
111	*PDGFRL*	rs4921799			1						1	Known
112	*PIK3R3*	rs499600			1						1	Novel
113	*PKHD1*	rs17752199	1								1	Known
114	*PLEKHA7*	rs4141194			1						1	Known
115	*PNPT1*	rs2627761	1								1	Known
116	*PRKAG2*	rs62478245	1								1	Known
117	*PRRX1*	rs76117299			1						1	Novel
118	*ROR1*	rs2806545			1						1	Novel
119	*RP11-259G18.1*	rs112073200				1					1	Novel
120	*SALL3*	rs145025615			1						1	Novel
121	*SALRNA1*	rs34935520	1								1	Known
122	*SEMA3C*	rs327636			1						1	Known
123	*SH2B3*	rs7137828		1							1	Known
124	*SIX1*	rs34935520			1						1	Known
125	*SIX4*	rs34935520			1						1	Known
126	*SLC2A12*	rs2627231			1						1	Novel
127	*SMG6*	rs2273984			1						1	Known
128	*SOCS3*	rs73382006			1						1	Novel
129	*STK38*	rs9380578			1						1	Known
130	*TBKBP1*	rs4794057								1	1	Known
131	*TCF12*	rs1647381			1						1	Known
132	*TGFBR3*	rs1192415	1								1	Known
133	*THSD4*	rs112786475			1						1	Novel
134	*TMEM136*	rs12806740						1			1	Known
135	*TMEM181*	rs4709212	1								1	Known
136	*TRIB2*	rs35002856			1						1	Novel
137	*TRIOBP*	rs5756813			1						1	Novel
138	*TSC22D2*	rs59500396			1						1	Known
139	*TTC28*	rs5752774			1						1	Known
140	*UNC5B*	rs79416409			1						1	Novel
141	*ZNF280D*	rs1647381	1								1	Known
142	*ZNF516*	rs72973714			1						1	Novel

POAG: primary open-angle glaucoma; nsSNP: non-synonymous single-nucleotide polymorphisms; DEPICT: data-driven expression-prioritized integration for complex traits; SMR: summary data-based Mendelian randomization; MSMR: SMR using mQTL data.; *** We assessed novelty using four searching approaches: (i) Phenoscanner database (http://www.phenoscanner.medschl.cam.ac.uk/, accessed on 20 February 2020); (ii) GWAS catalog (https://www.ebi.ac.uk/gwas/, accessed on 3 February 2020); (iii) PubMed database; (iv) Glaucoma (genetics) review papers: [10,11].

**Table 3 genes-13-01055-t003:** Significant MetaXcan results of POAG analysis using the DGN whole blood transcriptome model.

Gene	Chr	Position	Z_Score	Effect Size	*p*-Value	Var_g	pred_perf *r*^2^	pred_perf pval ^¥^	pred_perf qval	n_snps in Model	COLOC
*AFAP1*	4	7851047	−7.7319	−8.77 × 10^−2^	1.06 × 10^−14^	0.71242	0.72019	1.12 × 10^−256^	1.10 × 10^−254^	95	Passed
*CDKN2A*	9	21981525.5	−6.51	−3.07 × 10^−1^	7.51 × 10^−11^	0.04310	0.07153	1.45 × 10^−16^	3.79 × 10^−16^	19	Rejected
*TMEM136*	11	120200114.5	6.42912	7.36 × 10^−1^	1.28 × 10^−10^	0.00684	0.01375	3.59 × 10^−4^	5.59 × 10^−4^	12	Passed
*C22orf29*	22	19838040	6.10818	1.20 × 10^−1^	1.01 × 10^−9^	0.26323	0.30708	2.46 × 10^−75^	2.16 × 10^−74^	34	Passed
*FAM125B*	9	129179224	5.71259	1.32 × 10^−1^	1.11 × 10^−8^	0.17292	0.23350	4.06 × 10^−55^	2.54 × 10^−54^	48	Rejected
*EHBP1L1*	11	65351815	5.65037	5.88 × 10^−1^	1.60 × 10^−8^	0.00914	0.01596	1.20 × 10^−4^	1.92 × 10^−4^	10	Passed
*ACP2*	11	47265655	−5.6276	−8.67 × 10^−2^	1.83 × 10^−8^	0.35781	0.40112	1.57 × 10^−104^	2.19 × 10^−103^	30	Passed
*LTBP3*	11	65316338.5	−5.4319	−1.07 × 10^−1^	5.57 × 10^−8^	0.24976	0.26340	4.32 × 10^−63^	3.12 × 10^−62^	42	Passed
*GAS7*	17	9957897	−5.1001	−1.43 × 10^−1^	3.40 × 10^−7^	0.11391	0.10400	9.33 × 10^−24^	3.01 × 10^−23^	117	Rejected
*NR1H3*	11	47280123.5	−4.9212	−8.27 × 10^−2^	8.60 × 10^−7^	0.30798	0.28447	6.61 × 10^−69^	5.24 × 10^−68^	66	Passed
*C17orf69*	17	43711638	4.73445	7.05 × 10^−2^	2.20 × 10^−6^	0.44321	0.50460	1.77 × 10^−142^	4.25 × 10^−141^	28	Passed
*NPEPPS*	17	45650475	−4.6292	−1.86 × 10^−1^	3.67 × 10^−6^	0.05724	0.07549	1.97 × 10^−17^	5.27 × 10^−17^	22	Rejected

POAG: primary open-angle glaucoma; Chr: chromosome; SNP: single-nucleotide polymorphisms; DGN: Depression Gene Network; COLOC: colocalization test; Var_g: variance of gene expression explained by SNPs in the model; pred_perf_*r*^2^: correlation coefficient of tissue model’s correlation to gene’s measured transcriptome (prediction performance); pred_perf_*p*-value: *p*-value of tissue model’s correlation to gene’s measured transcriptome (prediction performance); pred_perf_q-value: q-value of tissue model’s correlation to gene’s measured transcriptome (prediction performance); n_snps_in_model: number of SNPs in the model; COLOC passed results are those with probability < 0.5 at hypothesis three and probability > 0.5 at hypothesis four; ^¥^ the Bonferroni corrected significance threshold for MetaXcan *p*-value and prediction performance *p*-value were 4.34 × 10^−6^ and 3.57 × 10^−3^, respectively.

**Table 4 genes-13-01055-t004:** Significant SMR results of POAG GWAS using eQTLGen dataset (https://www.eqtlgen.org/, accessed on 10 December 2019) of whole blood and [31].

Tissue	Gene Name	Top eQTL SNP	Top SNP Chr:Position	A1/A2	β GWAS	SE GWAS	P GWAS	β eQTL	SE eQTL	P eQTL	β SMR	SE SMR	P SMR	P HEIDI	HEIDI
**Blood**	*RP11-466F5.8*	rs2790049	1:165743523	A/G	0.3137	0.019	2.68 × 10^−61^	0.2014	0.0149	7.75 × 10^−42^	1.5574	0.1486	1.10 × 10^−25^	3.93 × 10^−5^	Rejected
	*RP11-217B7.2*	rs2980083	9:107691362	A/C	0.1039	0.0135	1.51 × 10^−14^	0.2011	0.0089	3.24 × 10^−114^	0.5166	0.071	3.32 × 10^−13^	5.72 × 10^−4^	Rejected
	*AFAP1*	rs62290601	4:7939008	A/T	−0.1024	0.0146	2.35 × 10^−12^	1.077	0.0091	0.00E+00 ^#^	−0.0951	0.0136	2.57 × 10^−12^	6.34 × 10^−12^	Rejected
	*MVB12B*	rs10122788	9:129206832	A/G	−0.0761	0.0135	1.52 × 10^−8^	−0.5181	0.0085	0.00E+00 ^#^	0.1469	0.0261	1.75 × 10^−8^	8.54 × 10^−1^	Passed
	*NR1H3*	rs326222	11:47259668	C/T	−0.0781	0.0143	4.32 × 10^−8^	0.3878	0.0086	0.00E+00 ^#^	−0.2013	0.037	5.40 × 10^−8^	1.39 × 10^−1^	Passed
	*LTBP3*	rs12789028	11:65326154	A/G	−0.0981	0.0181	5.97 × 10^−8^	0.4798	0.0118	0.00E+00 ^#^	−0.2045	0.0381	7.79 × 10^−8^	1.46 × 10^−1^	Passed
	*BICC1*	rs10740734	10:60364363	A/G	−0.0725	0.0134	6.78 × 10^−8^	0.1728	0.0089	3.58 × 10^−84^	−0.4196	0.0807	1.99 × 10^−7^	2.18 × 10^−1^	Passed
	*TMCO1*	rs4657473	1:165687151	T/C	−0.0941	0.0152	5.85 × 10^−10^	0.0872	0.0098	5.15 × 10^−19^	−1.08	0.2123	3.66 × 10^−7^	9.08 × 10^−6^	Rejected
	*EHBP1L1*	rs1346	11:65337251	T/A	−0.0979	0.018	5.06 × 10^−8^	−0.1473	0.0121	4.10 × 10^−34^	0.6648	0.1337	6.56 × 10^−7^	4.67 × 10^−1^	Passed
	*TXNRD2*	rs117985725	22:19860852	C/T	−0.133	0.0267	5.98 × 10^−7^	−0.4449	0.0209	1.76 × 10^−100^	0.299	0.0615	1.17 × 10^−6^	3.88 × 10^−6^	Rejected
	*ABCA1*	rs2487052	9:107686405	T/C	−0.0858	0.0169	3.61 × 10^−7^	−0.1588	0.0108	1.18 × 10^−48^	0.5402	0.1124	1.53 × 10^−6^	2.25 × 10^−2^	Passed
	*KANSL1-AS1*	rs112073200	17:44201791	C/G	0.0772	0.0163	2.22 × 10^−6^	0.9074	0.0128	0.00E+00 ^#^	0.0851	0.018	2.34 × 10^−6^	1.09 × 10^−2^	Passed
	*RP11-707O23.5*	rs112073200	17:44201791	C/G	0.0772	0.0163	2.22 × 10^−6^	0.8722	0.0129	0.00E+00 ^#^	0.0885	0.0188	2.35 × 10^−6^	2.34 × 10^−1^	Passed
	*RP11-259G18.1*	rs112073200	17:44201791	C/G	0.0772	0.0163	2.22 × 10^−6^	0.496	0.0137	1.90 × 10^−288^	0.1556	0.0332	2.70 × 10^−6^	5.19 × 10^−2^	Passed
	*DND1P1*	rs113991678	17:43795634	T/C	0.0766	0.0163	2.65 × 10^−6^	0.7525	0.013	0.00E+00 ^#^	0.1017	0.0217	2.85 × 10^−6^	1.72 × 10^−1^	Passed
	*CRHR1-IT1*	rs112560196	17:44200078	T/A	0.0764	0.0163	2.83 × 10^−6^	1.1528	0.0119	0.00E+00 ^#^	0.0663	0.0142	2.90 × 10^−6^	6.64 × 10^−1^	Passed
	*LRRC37A4P*	rs112560196	17:44200078	T/A	0.0764	0.0163	2.83 × 10^−6^	−0.9098	0.0127	0.00E+00 ^#^	−0.084	0.018	2.97 × 10^−6^	2.45 × 10^−1^	Passed
	*LRRC37A2*	rs112560196	17:44200078	T/A	0.0764	0.0163	2.83 × 10^−6^	0.7104	0.0132	0.00E+00 ^#^	0.1076	0.0231	3.08 × 10^−6^	7.72 × 10^−3^	Passed
**Brain**	*TXNRD2*	rs73148965	22:19872935	G/A	0.1463	0.0211	3.99 × 10^−12^	1.3077	0.0362	4.05 × 10^−285^	0.1119	0.0164	9.57 × 10^−12^	2.01 × 10^−2^	Passed
	*RP11-466F5.8*	rs10918274	1:165714416	C/T	−0.305	0.0188	5.96 × 10^−59^	−0.3408	0.0588	6.66 × 10^−9^	0.8949	0.1639	4.77 × 10^−8^	4.72 × 10^−3^	Rejected
	*CDKN2B-AS1*	rs504318	9:22024023	T/A	−0.1632	0.0135	1.50 × 10^−33^	−0.2904	0.0522	2.62 × 10^−8^	0.562	0.1112	4.33 × 10^−7^	2.03 × 10^−1^	Passed
	*CDKN2B*	rs490005	9:22020493	A/G	−0.1631	0.0135	1.24 × 10^−33^	−0.2126	0.0384	3.16 × 10^−8^	0.767	0.1525	4.90 × 10^−7^	1.41 × 10^−1^	Passed
	*RP11-217B7.2*	rs1800977	9:107690450	G/A	0.0846	0.0144	3.95 × 10^−9^	0.5958	0.073	3.31 × 10^−16^	0.142	0.0297	1.80 × 10^−6^	3.45 × 10^−3^	Rejected
	*RP11-707O23.5*	rs17575507	17:44134095	G/A	0.0771	0.0163	2.26 × 10^−6^	1.465	0.0381	0.00E+00 ^#^	0.0526	0.0112	2.68 × 10^−6^	5.56 × 10^−1^	Passed
	*LRRC37A2*	rs62641967	17:44047216	G/T	0.0766	0.0163	2.65 × 10^−6^	1.3074	0.0331	0.00E+00 ^#^	0.0586	0.0126	3.11 × 10^−6^	5.66 × 10^−2^	Passed
	*LRRC37A*	rs62641967	17:44047216	G/T	0.0766	0.0163	2.65 × 10^−6^	1.3199	0.0339	0.00E+00 ^#^	0.058	0.0124	3.13 × 10^−6^	5.31 × 10^−2^	Passed
	*LRRC37A4P*	rs112746008	17:44126650	T/C	0.077	0.0163	2.46 × 10^−6^	−1.3756	0.0444	4.34 × 10^−211^	−0.056	0.012	3.19 × 10^−6^	3.28 × 10^−1^	Passed
	*MAPT*	rs62641967	17:44047216	G/T	0.0766	0.0163	2.65 × 10^−6^	−1.1792	0.0341	2.77 × 10^−262^	−0.0649	0.014	3.26 × 10^−6^	5.28 × 10^−3^	Passed

POAG: primary open-angle glaucoma; SNP: single-nucleotide polymorphisms; SMR: summary-data-based Mendelian randomization; eQTL: expression quantitative trait loci; GWAS: genome-wide association study; HEIDI: heterogeneity independent instruments; b: effect size; se: standard error; p: *p*-value; A1: effect (risk) allele; A2: non-effect allele; **^#^**
*p*-values which are less than or equal to 3.27 × 10^−310^ are stored as 0.00E+00 due to arithmetic underflow condition.

**Table 5 genes-13-01055-t005:** Significant MSMR results of POAG GWAS using dataset of whole blood [36] and brain mQTLs [31].

Tissue	Probe ID	Nearest Gene	Top mQTL SNP	Top SNP Chr:Position	A1/A2	β GWAS	SE GWAS	P GWAS	β mQTL	SE mQTL	*P* mQTL	β MSMR	SE MSMR	P MSMR	P HEIDI
**Blood**	cg17332705	*AFAP1-AS1*	rs62290601	4:7939008	A/T	−0.1024	0.0146	2.35 × 10^−12^	0.2569	0.0330	6.68 × 10^−15^	−0.3986	0.0765	1.87 × 10^−7^	1.85 × 10^−3^
	cg24250820	*AFAP1*	rs55938116	4:7933940	A/C	−0.1036	0.0148	2.66 × 10^−12^	0.2685	0.0317	2.17 × 10^−17^	−0.3859	0.0715	6.78 × 10^−8^	1.08 × 10^−2^
	cg12728606	*AFAP1*	rs2891928	4:7924802	G/C	0.1443	0.0137	7.16 × 10^−26^	−0.5031	0.0329	6.38 × 10^−53^	−0.2869	0.0331	4.34 × 10^−18^	2.45 × 10^−3^
	cg24023194	*AFAP1*	rs56220381	4:7907636	A/G	−0.1294	0.0140	3.20 × 10^−20^	0.2220	0.0324	6.85 × 10^−12^	−0.5827	0.1059	3.75 × 10^−8^	1.56 × 10^−1^
	cg15957394	*AFAP1 (dist = 169)*	rs17771470	4:7931611	C/G	−0.1119	0.0143	5.39 × 10^−15^	−0.3150	0.0324	2.32 × 10^−22^	0.3552	0.0583	1.11 × 10^−9^	1.83 × 10^−3^
	cg19564367	*AFAP1 (dist = 198)*	rs56078222	4:7932073	C/T	−0.1037	0.0145	9.74 × 10^−13^	−0.2740	0.0350	4.96 × 10^−15^	0.3786	0.0718	1.34 × 10^−7^	1.57 × 10^−3^
	cg09806625	*EXOC2*	rs17135234	6:593109	C/A	0.1490	0.0190	4.04 × 10^−15^	1.4851	0.0317	0.00E+00 ^#^	0.1003	0.0130	9.52 × 10^−15^	1.84 × 10^−2^
	cg21084119	*EXOC2*	rs17135679	6:614787	C/T	0.1462	0.0186	3.43 × 10^−15^	−0.4399	0.0316	4.82 × 10^−44^	−0.3323	0.0485	7.20 × 10^−12^	1.28 × 10^−2^
	cg14812743	*PDE7B*	rs6570062	6:136388422	T/G	−0.0943	0.0139	1.32 × 10^−11^	−0.3324	0.0329	5.49 × 10^−24^	0.2837	0.0505	1.89 × 10^−8^	1.18 × 10^−2^
	cg14470647	*ABCA1*	rs1800977	9:107690450	A/G	−0.0846	0.0144	3.95 × 10^−9^	−0.3347	0.0330	3.60 × 10^−24^	0.2528	0.0497	3.56 × 10^−7^	2.43 × 10^−3^
	cg13430450	*ABCA1 (dist = 536)*	rs2422493	9:107690995	A/G	−0.1037	0.0136	2.06 × 10^−14^	−0.2364	0.0327	4.55 × 10^−13^	0.4389	0.0835	1.47 × 10^−7^	8.62 × 10^−3^
	cg05938607	*BICC1*	rs10740731	10:60348886	G/A	0.0736	0.0134	3.85 × 10^−8^	0.8782	0.0273	4.86 × 10^−227^	0.0838	0.0155	5.99 × 10^−8^	3.28 × 10^−1^
	cg12342675	*BICC1*	rs7474570	10:60343085	C/G	0.0714	0.0134	9.69 × 10^−8^	1.1076	0.0224	0.00E+00 ^#^	0.0645	0.0122	1.15 × 10^−7^	6.11 × 10^−1^
	cg10738003	*ARHGEF12*	rs7117321	11:120239051	C/T	0.0827	0.0135	9.00 × 10^−10^	−1.2781	0.0161	0.00E+00 ^#^	−0.0647	0.0106	1.01 × 10^−9^	7.19 × 10^−3^
**Brain**	cg15605172	*ACOXL*	rs6720503	2:111665137	A/G	0.0822	0.0142	6.79 × 10^−9^	1.2623	0.0341	1.00 × 10^−300^	0.0651	0.0114	1.02 × 10^−8^	1.55 × 10^−3^
	cg25107522	*DGKD*	rs7422272	2:234268308	A/C	−0.0759	0.0137	3.17 × 10^−8^	−1.2605	0.0340	1.00 × 10^−300^	0.0602	0.0110	4.47 × 10^−8^	2.31 × 10^−2^
	cg20312457	*AFAP1*	rs35609019	4:7847892	G/C	0.1148	0.0148	8.56 × 10^−15^	0.4871	0.0603	6.28 × 10^−16^	0.2356	0.0421	2.17 × 10^−8^	4.60 × 10^−3^
	cg24023194	*AFAP1*	rs13435730	4:7923991	G/A	0.1418	0.0137	4.81 × 10^−25^	0.4259	0.0625	9.67 × 10^−12^	0.3330	0.0585	1.29 × 10^−8^	6.70 × 10^−1^
	cg07406289	*AFAP1*	rs2385902	4:7921634	A/G	0.1419	0.0137	2.97 × 10^−25^	0.3745	0.0632	3.05 × 10^−9^	0.3789	0.0736	2.62 × 10^−7^	6.79 × 10^−1^
	cg08544002	*EXOC2*	rs17756712	6:625071	A/G	−0.1208	0.0169	9.46 × 10^−13^	−1.4534	0.0376	0.00E+00 ^#^	0.0831	0.0118	2.22 × 10^−12^	1.35 × 10^−2^
	cg21084119	*EXOC2*	rs17756712	6:625071	A/G	−0.1208	0.0169	9.46 × 10^−13^	−1.4224	0.0379	3.47 × 10^−308^	0.0849	0.0121	2.34 × 10^−12^	1.62 × 10^−3^
	cg14470647	*ABCA1*	rs2243312	9:107690124	A/G	0.0825	0.0143	8.49 × 10^−9^	0.5031	0.0440	3.10 × 10^−30^	0.1640	0.0319	2.71 × 10^−7^	6.13 × 10^−2^
	cg01159576	*LMX1B*	rs62578127	9:129386860	C/T	0.1279	0.0153	6.45 × 10^−17^	−0.3895	0.0477	3.33 × 10^−16^	−0.3282	0.0562	5.27 × 10^−9^	3.29 × 10^−2^
	cg09696939	*BICC1 (dist = 823)*	rs10740731	10:60348886	G/A	0.0736	0.0134	3.85 × 10^−8^	1.2146	0.0328	1.00 × 10^−300^	0.0606	0.0111	5.38 × 10^−8^	1.18 × 10^−2^
	cg00554250	*NR1H3*	rs1052373	11:47354787	C/T	−0.0843	0.0140	1.97 × 10^−9^	1.3202	0.0356	1.00 × 10^−300^	−0.0638	0.0108	3.15 × 10^−9^	3.34 × 10^−3^
	cg14056187	*C14orf39 (dist = 1445)*	rs7155448	14:60937851	C/T	−0.1184	0.0141	3.98 × 10^−17^	0.3808	0.0452	3.70 × 10^−17^	−0.3110	0.0522	2.64 × 10^−9^	2.22 × 10^−2^
	cg01170045	*SIX6 (dist = 4306)*	rs61991551	14:60982515	A/G	−0.1187	0.0141	3.46 × 10^−17^	−0.4162	0.0648	1.38 × 10^−10^	0.2851	0.0558	3.28 × 10^−7^	1.48 × 10^−1^
	cg10299448	*GAS7*	rs17687006	17:10019797	C/T	0.1102	0.0167	4.36 × 10^−11^	−0.6465	0.0533	8.03 × 10^−34^	−0.1705	0.0294	7.01 × 10^−9^	1.67 × 10^−2^
	cg15661389	*GAS7*	rs17687006	17:10019797	C/T	0.1102	0.0167	4.36 × 10^−11^	−0.5267	0.0548	6.75 × 10^−22^	−0.2092	0.0385	5.42 × 10^−8^	3.14 × 10^−2^
	cg19784903	*TBKBP1*	rs4794057	17:45786452	C/T	−0.0692	0.0137	4.14 × 10^−7^	1.0926	0.0277	0.00E+00 ^#^	−0.0634	0.0126	5.14 × 10^−7^	2.93 × 10^−3^

POAG: primary open-angle glaucoma; SNP: single-nucleotide polymorphisms; SMR: summary-data-based Mendelian randomization; eQTL: expression quantitative trait loci; GWAS: genome-wide association study; HEIDI: heterogeneity in dependent instruments; b: effect size; se: standard error; p: *p*-value; A1: effect (risk) allele; A2: non-effect allele; P GWAS: *p*-value of SNP association with POAG; ^#^ *p*-values which are less than or equal to 3.27 × 10^−310^ are stored as 0.00E+00 due to arithmetic underflow condition. Only HEIDI-passed results are shown: pHEIDI ≥ 2.78 × 10^−3^ (0.05/18 SMR significant probes) in blood and pHEIDI ≥ 5.0 × 10^−3^ (0.05/10 SMR significant probes) in the brain.

## Data Availability

The POAG GWAS dataset analyzed during the current study is available in the GWAS catalog repository (http://ftp.ebi.ac.uk/pub/databases/gwas/summary_statistics/ under Study accession GCST006065, accessed on 21 April 2019). Whole blood and brain eQTL datasets for TWAS analyses are available from eQTLGen website (https://www.eqtlgen.org/, accessed on 10 December 2019) and SMR data resource (https://cnsgenomics.com/software/smr/, accessed on 19 December 2019), respectively. Gene expression data for ocular tissues are available at Ocular Tissue DataBase (OTDB) (https://genome.uiowa.edu/otdb/, accessed on 23 January 2020). All results generated during this study are included in this published article and its Appendix A.

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
