# Peer review of "Bioinformatic Prioritization and Functional Annotation of GWAS-Based Candidate Genes for Primary Open-Angle Glaucoma"

_genes, 2022, doi:10.3390/genes13061055_

Round 1
Reviewer 1 Report
Post-GWAS analysis is a powerful approach to understand disease mechanisms and gene regulatory networks. In this manuscript, Asefa et al. combined and analyzed in silico sequencing, POAG methylation datasets, and POAG gene-expression datasets. They identified 142 genes that could be the most causal or relevant genes for POAG, 64 of which were novel. The authors also performed functional assessment analysis to illustrate the molecular mechanisms underlying glaucoma pathogenesis. While the authors have made an amount of effort for data analysis, there are some points that need to be adequately addressed.
Major points:
1. How many of identified POAG associated variants are located in non-coding regions? And how these SNPs influence gene expression regulation.
2. Could the authors use a regional plot to show the regional visualization of genome-wide association scan results for POAG, for example: using LocusZoom software. It would be helpful for readers to better understand the data.
3. Mostly single SNPs have a small effect and have limited predictive power. Could the authors also perform polygenic risk score (PRS) analysis.
4. How far away from a SNP/CpG site was considered to be positively associated with its target gene?
5. Are there any CpG-SNPs that were identified in POAG, or SNPs close to the CpG sites?
Minor points:
1. Line 292, 323, 458, and line 476 contained double space, please correct them.
2. Line 749-750, “This section is not mandatory but may be added if there are patents resulting from the work reported in this manuscript” should be deleted.
Author Response
Our responses to reviewers #1 and #2 are attached as a word file.

Reviewer 2 Report
Manuscript entitled „Bioinformatic Prioritization and Functional Annotation of GWAS-Based Candidate Genes for Primary Open-Angle Glaucoma” presents the analysis of genes that may be involved in primary open-angle glaucoma pathogenesis.
In my opinion the manuscript is well written, informative. Interesting tables and figures make the data more clear.
My comments/suggestions:
Abstract – In my opinion the sentence „This study aims to prioritize likely causal genes and describe their functional characteristics and relationship” should be clarified according to the subject of manuscript.
Fig. 4 – make the graph more visible, please
Line 513: „with IOP,[53]” – is this correct?
748-749: „This section is not mandatory but may be added if there are patents resulting from the work 749 reported in this manuscript” – delete it, please
Author Response

(The authors gave the same response as above.)
